# Radical transformation pathway towards sustainable electricity via evolutionary steps

Dmitrii Bogdanov [1], Javier Farfan[1], Kristina Sadovskaia[1], Arman Aghahosseini [1], Michael Child [1], Ashish Gulagi[1], Ayobami Solomon Oyewo[1], Larissa de Souza Noel Simas Barbosa[2] & Christian Breyer [1]

A transition towards long-term sustainability in global energy systems based on renewable energy resources can mitigate several growing threats to human society simultaneously: greenhouse gas emissions, human-induced climate deviations, and the exceeding of critical planetary boundaries. However, the optimal structure of future systems and potential transition pathways are still open questions. This research describes a global, 100% renewable electricity system, which can be achieved by 2050, and the steps required to enable a realistic transition that prevents societal disruption. Modelling results show that a carbon neutral electricity system can be built in all regions of the world in an economically feasible manner. This radical transformation will require steady but evolutionary changes for the next 35 years, and will lead to sustainable and affordable power supply globally.

[1] Lappeenranta University of Technology, Skinnarilankatu 34, FI-53850 Lappeenranta, Finland. [2] Luiz de Queiroz College of Agriculture, University of São Paulo, Piracicaba, Brazil. Correspondence and requests for materials should be addressed to D.B. (email: Dmitrii.Bogdanov@lut.fi) or to C.B. (email: Christian.Breyer@lut.fi)

Several milestones have recently been reached that are indicative of growing environment risk: average global temperature[1], greenhouse gas (GHG) concentrations, and GHG emission levels[2] have all hit highs for the industrial era. Further, there are increasing reports of climate deviations around the globe[3], and coral reefs represent the first major planetary ecosystem under threat of major collapse[4–6]. It has become impossible to ignore the challenge of climate change given the magnitude of evidence, and society is more focused on climate change mitigation. The Paris Agreement[7] was an important first step towards united energy policy[4]. Fossil fuel-related GHG emissions were recognized as a major cause of global warming, a key characteristic of the Anthropocene era[8] and a major threat to the future of civilization. Global society and its leaders recognize the need for a transition towards sustainable energy systems in order to limit climate change and guarantee future development[9]. This awareness has resulted in increased interest in and focus on renewable energy (RE), further accelerated by the latest IPCC SR1.5 report[10]. And increasing numbers of energy scenarios consider RE as a major part of the energy system in the decades to come[11–21]. While the International Energy Agency (IEA) shows vision inertia and constantly underestimates the role of RE in its World Energy Outlook (WEO) scenarios[10], as discussed in Creutzig et al.[15]. Other organizations are more visionary. Greenpeace shows much higher reliance on RE in its Advanced [r]evolution scenario[12,13]. Based on the historical impact of decreasing costs and rapidly increasing installations, Haegel et al.[14], Creutzig et al.[15] and Pursiheimo et al.[16] expect solar photovoltaics (PV) to emerge as a main source of electricity in the future with terawatt (TW) scale installed capacities[17], and others ponder the role of RE in their scenarios[4]. Lastly, 100% RE-based energy systems are discussed as a feasible solution in different regions of the world and globally, as listed by Brown et al.[18]. Further, Jacobson et al.[19] reported the possibility of satisfying global energy demand with only renewable energy, while Breyer et al.[20] showed in hourly resolution that electricity supply based fully on RE is possible, for attractive cost, and for all regions globally for 2030 assumptions. The International Renewable Energy Agency is the first international governmental institution which confirms that electricity supply very close to 100% RE can be expected for major countries and economic rims in 2050, in particular China, EU, and India[22].

Thus, currently available generation and storage technologies are sufficient for nearly 100% power system operation. Available RE energy resources are adequate to satisfy current and future power sector demand in every region of the world[20]. The remaining challenges are the stability of an energy system with a low share of rotating generation machinery and the societal acceptance of the RE technologies. An RE-based system will have lower physical inertia and will not be able to mitigate a short-term imbalance of generation and demand. However, a lack of physical inertia in a system with a high RE share can be overcome with the integration of synthetic inertia, essentially improved algorithms of power converters of RE generation and storage capacities[23]. A recent synthetic inertia investigation for a 100% renewable power system for sub-Saharan Africa confirmed the attractiveness of this approach[24]. Raw material scarcities can be limiting factors for some technologies in the future, as lithium for lithium batteries, or dysprosium and neodymium for wind turbines with permanent magnet drives. However, in all these fields alternative technologies exist, using alternative raw materials (i.e. non-lithium ion batteries[25], electrically excited synchronous generators and others in wind turbines[26]). For silicon-based PV, representing more than 95% of the annually added solar PV capacity, the main raw materials from a mass content point of view are silicon (for glass and semiconductor material) and aluminum, two of most abundant elements in the Earth's crust. Mass content of doping materials is negligible. Silicon solar cells often use silver, but this is not mandatory, as documented by the high-efficiency PV cells of SunPower. In total, there is no material limitation known to produce these capacities of PV.

Societal acceptance is a more uncertain aspect. In our work we assume that up to 6% of regional area can be used for PV system installations, 4% of area can be used for wind farm installations, hydro generation capacities can be increased at most by 50%. The latter is mainly related to the commissioning of under construction capacities and repowering of old hydropower plants. Social acceptance of technologies varies over time and cannot be derived or estimated by techno-economic analysis. All major concerns about the technical feasibility and economic viability of 100% renewable systems, which still persist, are summarized by Brown et al.[18].

The aforementioned scenarios are fully or partly limited in temporal resolution, spatial resolution, speed of defossilization, energy transition pathway description, cost efficiency, and technological scope. Therefore, a new methodology was needed that overcomes these limitations.

Accordingly, a simulation is carried out on a global scale using the LUT Energy System Transition model. The world is structured into nine major regions: Europe, Eurasia, Middle East and North Africa (MENA), sub-Saharan Africa (SSA), South Asian Association for Regional Cooperation (SAARC), Northeast Asia, Southeast Asia and the Pacific Rim, North America, and South America. In total, the world is divided into 145 subregions (Supplementary Table 1), balanced to represent comparable shares of global power demand, population and land area. Both hourly resolution and the regional structure are considered to avoid underestimating RE source variability.

The modeled transition starts from the existing power system structure as of 2015, and existing capacities are decommissioned only after reaching their technical lifetimes[27]. The speed of RE capacity deployment is limited to avoid an unrealistically fast transition and is based on empirical data[27]. For each transition step, linear optimization of the power system is performed, with a target of minimum annualized system cost under given constraints. The annual cost includes annualized capital expenditures (capex), operational expenditures, ramping costs for each technology, fuel costs, and GHG emission costs. The final step of the transition process is to reach a 100% sustainable and carbon neutral energy system, independent of fossil and nuclear fuel supply. Nuclear energy is not considered as sustainable energy in this analysis due to high societal risk, unsolved radioactive waste problems, and substantial economic issues[28,29]. However, existing plants are operated until the end of their technical lifetimes. Contrary to other scenarios[30], it is shown that nuclear energy is unnecessary for effective climate change mitigation.

## Results

**Existing power sector and RE potential.** Fossil fuels are the backbone of the present global energy system, contributing to 65% of all electricity generated[11]. Most existing RE is generated by hydropower (16%), while solar PV (1.2%) and wind energy (4%) contribute less[11]. However, solar PV and wind energy show high compound annual capacity growth rates of 48% and 21% for the period 2006−2016, and 33% and 12.5% in 2016[31], respectively, and their high technical potentials of 87.5–2770 PWh$_{el}$ (solar PV) and 23.6–161 PWh$_{el}$ (wind energy)[32] are distributed over the planet much more evenly than hydropower or fossil resources. Still, some regions have better wind conditions, some excellent solar irradiation, and some benefit from available hydropower potential or substantial biomass resources. Every

region has unique climatic conditions and RE potentials, which will lead to specific optimal structures of respective 100% RE systems.

The energy transition will depend not only on RE resource conditions, but also on how various RE sources complement each other in different regions. Some regions, like MENA, have an excellent and stable solar resource, which will lead to high shares of solar PV, likewise for all Sun Belt countries. Eurasia has a harsh continental climate with cold winters, during which electricity demand strongly increases while PV generation decreases. Meanwhile, wide plains of Eurasia are ideal for wind energy generation; high wind speeds lead to low generation cost, while low population density enables the installation of large-scale capacities. Europe, despite its small size, includes highly different regions: windy Britain, Norway with abundant hydropower potential, the sunny Iberian Peninsula and Balkans, and most other countries with a mix of these extremes. Regional descriptions, data on RE resources potentials applied in this research, installed capacity limits for RE and the projected power demand for all 145 regions are presented in Supplementary Tables 1–4, respectively.

Existing capacity structures also vary globally. Some regions rely mostly on coal capacities (e.g. Poland, Kazakhstan, India, Mongolia), which lead to very high GHG emissions. Others mainly rely on gas generation (e.g. Argentina, Belarus, Egypt, Algeria). Some countries have already integrated significant capacities of PV and wind into their power systems (e.g. Italy, Spain, Germany, Denmark, Uruguay), and some have built substantial hydropower capacities (e.g. Norway, Iceland, Myanmar, Laos). By the age structure of installed capacities, regions can be divided into two: first, regions with growing installation rates of new power generation capacity, and second, regions where maximum installation rates have already been surpassed. In Europe and Eurasia, the peak of capacity growth has already passed, and the share of gas-based electricity generation is high. On the other hand, Northeast Asia and the SAARC region have coal-based power supply with fast growing capacities. Recently, RE capacity shares have grown rapidly[27,31]. However, huge coal capacities installed in recent decades will burden the transition process.

The transition process will depend on many parameters, such as regional economic situations, social acceptance of fossil fuels, nuclear energy and renewables, and political concerns[33], but most importantly on future electricity generation costs. Financial and technical assumptions for all applied technologies and data sources are presented in Supplementary Tables 5–8 in the Supplementary Material. The cost assumptions of RE technologies consider major trends in learning curves and increasing adoption rates that have a huge impact on future scenarios. The falling costs of renewable electricity generation and supporting storage technologies will be the driving force of the energy transition: solar PV has already become the least cost energy source in many regions of the world[30], and this decline is expected to continue[14]. Continued storage cost decrease[34,35] will make 100% renewable electricity systems highly cost competitive.

The modeling was performed using the LUT Energy System Transition model. Future electricity consumption assumptions are based on IEA estimations[36] and represent the development of the existing power sector without consideration of possible additional electricity demand due to massive electrification of heat and transport sectors, as discussed for the case of Europe[37]. Solar and wind resource assumptions are based on a NASA database and recalculated for the case of currently widely available RE generation technologies (PV with 15% efficiency and Enercon E-101 turbines). Further details on available RE resources, power demand, technical and financial assumptions,

for all observed technologies, are represented in Supplementary Tables 2–8.

**Future uncertainty**. All the parameters influencing the future system development and energy cost are uncertain including political will, societal acceptance, and the cost of energy system elements. With the techno-economic approach, we assume that political and societal will follow the common good: low-cost and sustainable energy supply. Cost assumptions for the technologies are based on a set of reliable sources, as presented in the Supplementary Material. However, we also apply a ±10% cost range for the most important generation technologies: solar PV and wind power plants, since the cost development for these is well studied. For most important storage technologies: battery storage and power-to-gas system elements (electrolyzers, $CO_2$ direct air capture and methanation units), we assume a wider range of ±30%, since these technologies have not yet reached technical maturity.

Other aspects are unforeseen costs and cost overruns. Sovacool et al.[38] show that hydropower plants and nuclear reactors have the highest probability and magnitude of cost overruns (71% and 117% cost increase, respectively), much higher than for thermal power plants (13%). Cost overruns for modern renewables are much lower: 8% for wind power plants and 1% for solar PV power plants. For power storage technologies, such statistics are unavailable so we assume 10% cost escalation for power storage projects. In total, cost overruns of the system can reach 6% in 2050, weighted according to the mix of technologies.

A major factor of uncertainty can be the cost of capital, which is set for this research to a uniform weighted average cost of capital of 7%. This can deviate to higher values reflecting higher risk, but also to lower values. The latter has been recently observed for the case of solar PV and wind power plant investments in Germany, which have been reported for weighted average cost of capital of 2.5% and 2.75%[39], respectively, assuming a standard 30% equity and 70% debt financing.

For simplicity, cost diagrams are given for the median costs of technologies presented in the Supplementary Material and without additional unforeseeable costs.

**Transformation towards 100% renewable electricity**. Modeling results show that a 100% carbon neutral RE-based electricity system is possible by 2050. Such an energy system is economically feasible, at a levelized cost of electricity (LCOE) of 52 €/MWh (uncertainty range 45–58 €/MWh), less than the present 70 €/MWh. Solar PV will be the main source of electricity, generating almost 70% of all electricity, and wind nearly 18%. Diverse RE resource availability and starting system configurations will result in different system transitions. Modeled regional energy systems are classified into four groups (see global overview in Fig. 1). Shares of solar PV, wind turbines and power plants in total electricity generation during transition is shown in Supplementary Figs. 1–3. Each of the 145 systems is unique, even the systems of the same type still have substantial differences. For instance, India and Saudi Arabia are both located in the Sun Belt and have PV-based energy systems; however, Indian monsoons will increase the mid-term share of wind and storage technologies[40] compared to Saudi Arabia, which has a more stable solar resource[41].

Results show the global generation capacities in 2050 will exceed 28 TW, of which 22.0 TW will be solar PV and 3.2 TW will be wind turbines, representing about 39,130 TWh and 10,160 TWh of solar PV and wind electricity generation. Accordingly, solar PV capacity increases by about 100 times compared to 2015, and wind energy capacity by about 8 times. Achieving this will be

challenging but manageable[14]. In 2030, the global generation capacities for solar PV will be around 7 TW, which is within the expectation of Haegel et al.[14] and consistent with recent actual installation growth rates, whereas the solar PV generation in 2050 is very close to the results of Creutzig et al.[15], who consider the full energy system. Hydropower capacities will not grow that significantly, only about 25%, which mostly represents commissioning of current under construction capacities (18%) and repowering and modernization of existing hydropower plants, mainly due to the limited potential of unexploited hydro resources, negative impacts of large-scale hydropower projects[28,42] and decreasing competitiveness to solar PV and wind energy. Contributions of other generation technologies, bioenergy and geothermal generation may be not significant on a global scale, but still important for some regions.

The other major structural change in the system is the role of storage, which becomes an inevitable element of the power system, supplying 31% of total electricity demand. The most important role will be played by battery storage, which complements the major energy source, solar PV. Diurnal Li-ion battery storage will be most important both from throughput and

power capacity perspectives. Battery storage will reach about 8 TW power capacity and 48 TWh$_{cap}$ of energy storage capacity, but seasonal gas storage will be the largest from a storage capacity perspective. About 1000 TWh$_{cap}$ of gas storage capacity will be needed to compensate seasonal demand and generation fluctuations in high latitudes, which is comparable to the current gas storage capacity in Europe. On average, gas storage is used equally for storing biomethane from biomass sources and synthetic methane produced by power-to-gas units[43].

The range of LCOE for countries will be 27–70 €/MWh around a global average of 52 €/MWh (uncertainty range 45–58 €/MWh) for 2050. The lowest LCOE is reached in Iceland, a country with excellent geothermal energy and hydropower potential. The highest LCOE is recorded for Belarus, a country with moderate solar irradiation, moderate wind resources and limited hydropower potential. The global average cost of electricity generation in 2050 will be about 25% lower than for 2015. Moreover, after 2050 the cost will continue to decline a further 20% due to reinvestments in RE capacities, which saw cost declines during the transition. A global overview of LCOE by country is depicted in Fig. 2.

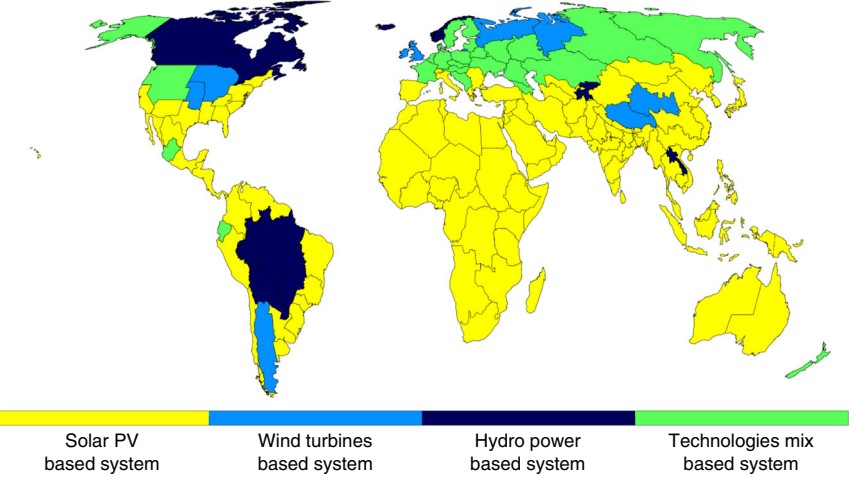

**Fig. 1** Main types of 100% renewable electricity systems. Four main types of RE-based power systems are identified based on their main source of electricity (more than 50% share of electricity generation). If none of the technologies have a share exceeding 50%, then the type is defined as "Technology mix-based system"

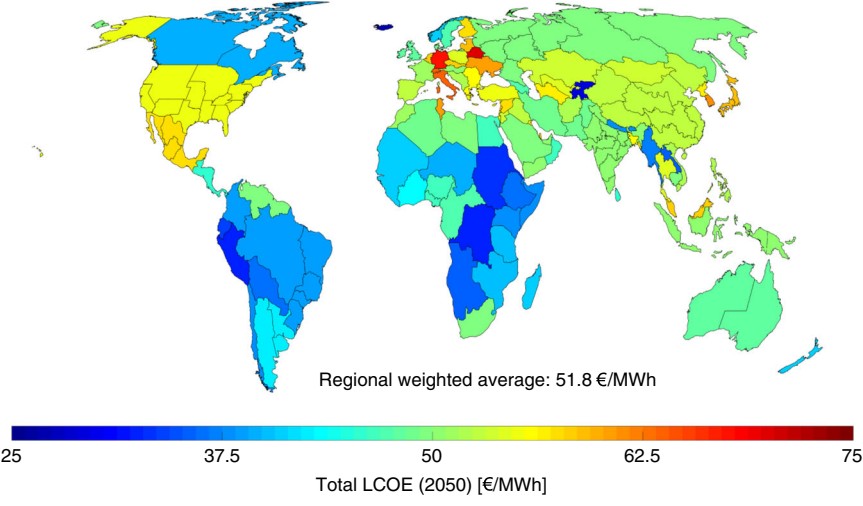

**Fig. 2** Levelized cost of electricity for 100% renewable electricity systems in 2050. Country average numbers are presented. Numbers are calculated based on the generation mix for 2050 and financial and technical assumptions for all electricity system components. For countries divided in several regions, levelized cost of electricity is calculated as weighted average

The countries with dominance of dispatchable RE generation, like Iceland with hydropower and geothermal generation, or Tajikistan and Kyrgyzstan with high shares of hydro reservoirs, will have the lowest electricity generation cost globally. However, this does not mean dispatchable RE generation is a condition to have low LCOE in an RE-based system. For the year 2050, LCOE is very low in Brazil and equatorial countries with energy systems based on a mix of various energy sources. In these countries, the power system must include a well-developed transmission and distribution grid to provide access to the mix of resources distributed throughout the countries. Very low-cost levels for 100% renewable electricity can be reached in completely different conditions. Kenya, Uganda, Somalia, and Djibouti have limited access to dispatchable RE resources, and would have power systems based on variable RE sources, mostly solar PV. However, the total LCOE is low, at 35 €/MWh (uncertainty range 30–40 €/MWh). Climate conditions in such countries complement solar-based systems. Low seasonal demand fluctuation and an optimal diurnal solar cycle result in low long-term storage demand, so electricity can be supplied by mainly solar PV and limited battery storage.

Some developed countries, such as Germany, Italy, Switzerland, Japan, and Korea, have significantly higher LCOE than their neighbors. One of the reasons is the high activity of PV prosumers in these regions. PV prosumers generate electricity at higher cost, but it is still cheaper than buying electricity from distribution companies. At the same time, PV prosumers hardly buy electricity during peak production, which increases demand for storage and finally storage costs of the system. For Korea and Japan, high cost is also driven by very high population density, which limits deployment of area intensive wind energy. Very high electricity demand and limited area result in an energy resource mix that leads to higher cost of electricity compared to areas with lower population density. These issues may be solved with additional, progressive regulations of the prosumers in the first case, and higher social acceptance of renewables, in particular wind energy, in the future. This will enable decreasing electricity cost in some regions.

**Radical transition in evolutionary steps**. The transition towards a 100% renewable electricity system will demand radical changes in system structure. Technology and generation mixes will change drastically, while a new storage sector will emerge. At first, wind energy and solar PV capacities grow at similar rates. In most energy-intensive regions wind generation is the cheapest source of electricity for the first 5-year steps of the transition, while expensive storage limits PV integration. The ongoing cost decline of PV systems and battery storage makes PV substantially more competitive than wind energy in many regions. Particularly in the Sun Belt, this leads to growth stagnation of wind capacities beyond 2030 and most new capacities installed are PV.

Biomass and biogas are very valuable resources for the system through the whole transition period; however, their impacts are rather small because of limited sustainable biomass resources and the rather high cost of solid biomass resources[44,45]. We assume these to be on the level of about 1900 TWh for biogas and 6400 TWh for solid biomass residues and wastes. During the first steps of transition, biomass and biogas are used for baseload electricity generation. Later, as the growing share of RE generation results in an increased need for system flexibility, biomass capacities start to play a regulatory role, and biogas is converted to biomethane and stored in gas storage. Finally, biomethane and solid biomass show their highest value as dispatchable renewable energy sources. In 2050 all biogas is used for electricity generation, while only one-third of available solid biomass is used globally, mostly in the

regions with high seasonality of RE resources and electricity demand.

Storage technologies emerge from very low levels to provide more than 15,000 $TWh_{el}$ in 2050. Gas storage operates as seasonal storage and emerges quite early to store biomethane for gas turbines. At later transition stages SNG is also stored in the same storage due to the same chemical identity. Gas storage is highly important for countries with strong seasonal variations in generation and demand. For other countries, in particular in the Sun Belt, diurnal battery storage is far more important as it supports the PV-based system. Battery storage emerges around 2030, when the PV capacity share reaches 50%. Beyond 2030, battery capacity steadily grows with PV generation. Shares of batteries in the total power supply trough the transition are presented in Supplementary Fig. 4. The total electricity throughput of battery storage, however, is much higher than for gas storage, since batteries are operated daily, leading to around 300 full charge cycles per year, instead of less than two for gas storage due to seasonal discharge. The structure of the power capacities, generation, storage capacities and storage throughput for each 5-year step are presented in Fig. 3. Installed capacities and generation structure through the transition for the world and all major regions are presented in Supplementary Figs. 5–13 and numerically in Supplementary Tables 9–28.

The very high share of solar PV of about 70% in total electricity generation in the year 2050 implies a consideration in potential limitations. The solar resource is not limited since only a small fraction of total available solar resources are used. As well, only a relatively small amount of land is needed, thereof a considerable amount in zero impact areas, such as rooftops. Energetic sustainability is given since the energy payback time for newly installed systems is about 1 year in global average resource conditions[46] and expected to further decline, in particular due to the energetic learning curve for solar PV systems[47]. Fundamental material limitations are not known, since the major input materials are $SiO_2$ for glass and silicon, and aluminum and hydrocarbons for foils. Silver is used for charge carrier extraction in some PV technologies, but could be substituted by copper-based solutions. The industrial manufacturing capacities can be ramped up more quickly as markets grow[48], which is a major reason for the continued steep cost decline, and industrial proof that fast growing markets can be served.

The most challenging period of the transition is the 2020s. As very large fossil capacities are decommissioned, they are substituted by renewables. However, in Eurasia existing capacities are very old, since most were built before 1990, and complete substitution of these capacities with renewables so quickly is unlikely. In such regions, additional capacities of gas turbines are installed to balance supply and demand. Later these gas turbines become part of seasonal storage and use carbon neutral biomethane or SNG as fuel, providing 2% of global demand in 2050. The 2030s and 2040s see a more gradual transition, since decommissioned fossil capacities are substituted by new RE capacities, and the first large-scale RE reinvestments happen. Most defossilization happens before 2030, while the assumed GHG emissions price is low and should not have significant impact on the cost. Until then, old and inefficient coal capacities are retired, substituted by RE generation, and the fossil capacity role changes from more baseload power generation to auxiliary generation. Later GHG emissions price increases and the role of fossil-based generation proceeds to decline. By 2035, GHG emissions can shrink by 90% compared to 2015. Beyond 2035, the system evolves to cover the growing electricity demand in developing and emerging countries and reach even higher defossilisation levels. The final 5-year step to a sustainable and carbon neutral system is demanding, as the last amounts of fossil

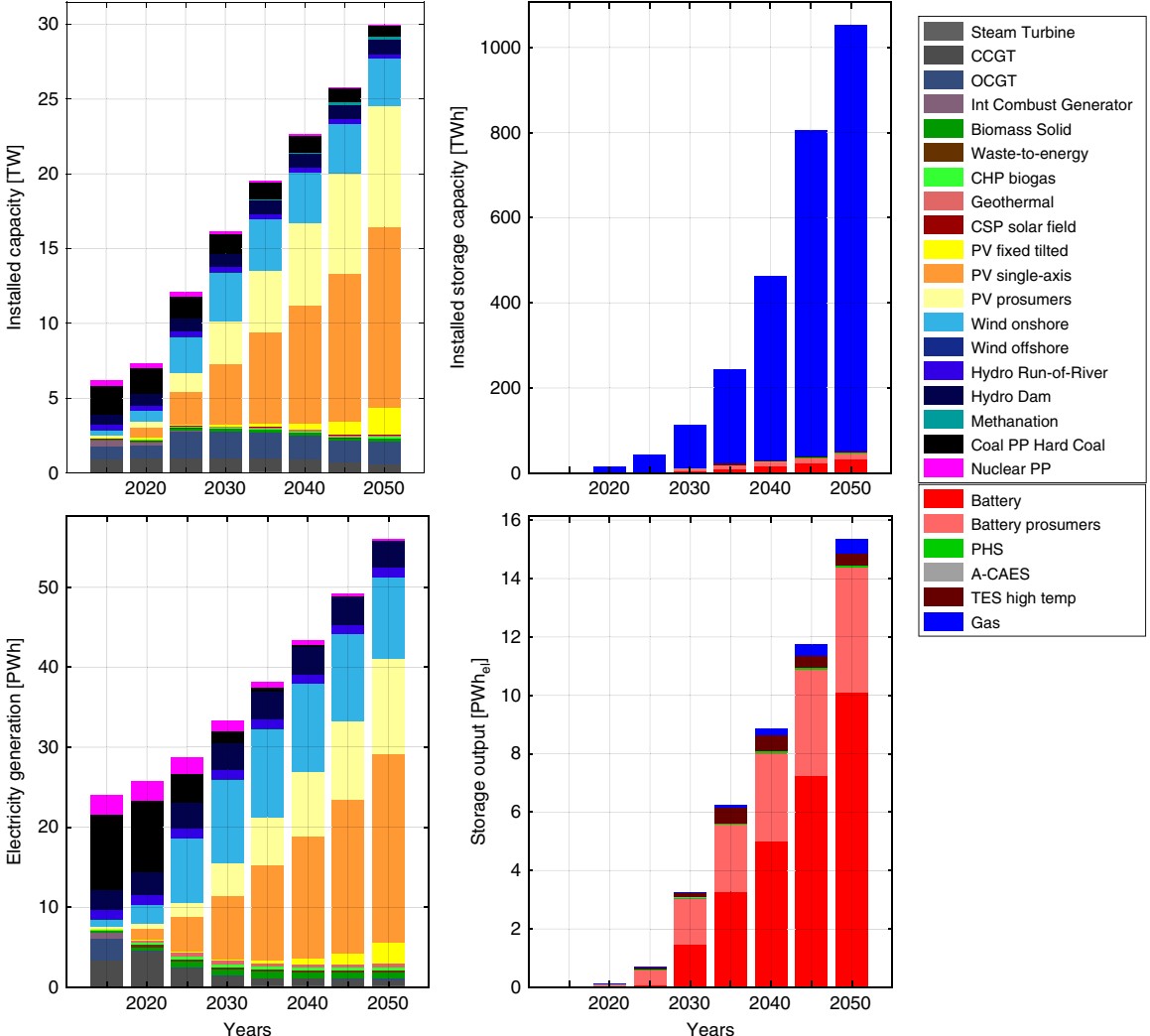

**Fig. 3** Power and storage capacities, power generation and storage throughput from 2015 to 2050. During the first steps of the transition most of new installed capacities are represented by wind turbines, as the least cost source of electricity during this time in most regions. Later with cost decline of PV and battery storage technologies, and utilization of most efficient wind generation sites, the share of PV in new installed capacities becomes dominant. Some wind turbine capacities are reinstalled in the later periods of the transition to substitute decommissioned old turbines. Overall growth of cumulated installed capacities is initiated by both growth of the power demand and the generally lower FLh of RE sources. Substantial growth of storage technologies capacities starts after 2030, when the VRE generation share exceeds 50% in most of the regions. PV photovoltaics, RE renewable energy

fuels are the most challenging to substitute. The global GHG emissions for each 5-year step are presented in Fig. 4. GHG emissions for all major regions are presented in Supplementary Figs. 14–22 and Supplementary Table 29. The total LCOE decreases with growth in the RE share, which implies that storage extra cost is well compensated by the very low cost of renewable electricity generation. After a small increase in LCOE in the years 2025–2030 related to the integration of RE capacities, total LCOE decreases due to continued development of RE technologies and related RE capital expenditure reduction. This trend is observed globally. Significant decrease of transmission and distribution grid losses expected in developing countries[49] also lead to LCOE decrease. The global LCOE breakdown for each 5-year step is presented in Fig. 5.

During the transition, the electricity cost structure changes drastically. Initially, half of the system LCOE refers to capex of the generation (LCOE primary), one-third to fuel cost and the rest to interregional power transmission (LCOT), curtailment losses (LCOC) and to lower levels of GHG emission cost. The share of fuel cost decreases and becomes negligible after 2035,

while the storage cost (LCOS) share grows due to increasing storage. At the same time, the share of capital and fixed operational expenditures in LCOE increases with the integration of higher shares of RE generation technologies, which have almost no fuel cost in comparison to traditional fossil-based generation. Major regions' LCOE breakdown for each 5-year step is presented in Supplementary Figs. 23–41.

**Investments during the transition.** The transition of the power sector will demand high capital expenditures (capex) of around 22.5 trillion € (uncertainty range 19–25.5 trillion €), or on average about 650 b€ per year. This is comparable to current investments in power generation, power transmission and fossil fuels for use in the power sector. In addition, it is significantly lower than the total energy system investment of 1308 b€ and global electricity investments of 552 b€ in 2016[51]. Capex in the power sector for all major regions for each 5-year step is presented in Table 1. The 2020s are the most challenging period due to a peak in old power capacity decommissioning. Lifetime extensions of old and

inefficient fossil generation would seriously violate GHG emission limits and must not be allowed. All these capacities will be substituted by RE capacities or, in most extreme cases for a short intermediate period, by gas-fueled gas turbines. Capital expenditures breakdown by technologies for all major regions are presented in Supplementary Figs. 42–51.

During the 2020s capex spikes to about 900 b€ a year, and later stabilizes at about 600 b€ per year. However, the situation widely depends on the region and past energy policy. Additional costs due to cost overruns at the system average level of about 6% may have to be also considered. Regional transmission and distribution grid reinforcements may increase total capital expenditures by 10–15% dependent on the grid structure and level of demand centralization[17]. Regions with high shares of pre-1990, fossil-based capacities face the biggest challenges. Investment demand spikes in Eurasia and North America, with the highest share of lifetime extended capacities, while in Europe or South America the transition can be more balanced. Moreover, the consequences

of past policy failures persist in the system even after 2030, as very large capacities installed in 2020 must be reinvested in 2050 and these waves of reinvestments remain for long periods. So, a late start of the energy transition and extension of business as usual policies will result in continued challenges in future. The transition would need to be even faster, and demand extra investments while conventional assets will most likely become stranded. The distorted investment cycle will remain longer. The system transition must be accomplished in the most optimal way, which will allow a fast but gradual evolution towards 100% renewables without major disruptions.

**Energy system models towards higher share of renewables.** Jacobson et al.[19], Sgouridis et al.[21], Löffler et al.[52], Pursiheimo et al.[16] and Teske et al.[12,13] also confirm that the global transformation towards RE-based systems is possible and affordable in economic and energetic terms. Different modeling approaches result in different shares of generation technologies in the global mix and GHG emission reduction trends, but they commonly recognize solar, wind and hydro as the most important energy sources. However, hourly resolution, latest cost trends, and an explicit focus on energy storage technologies, applied in this research, led to a much lower share of concentrating solar thermal power (CSP) plants and a higher relevance of solar PV. Integration of an electrified transport sector, electrical heating and cooling demand, and demand side management would help the system to accommodate even higher shares of PV[14]. This is confirmed by Pursiheimo et al.[16], who found an even higher solar PV share than in this study. However, due to the limited temporal resolution of the model used, they suggest carrying out detailed studies in higher temporal resolution to reduce uncertainties, which is the methodological core of this research. Jacobson et al.[19] show that 100% RE systems will positively impact society with several co-benefits: defossilization resulting in lower GHG emissions, heavy metal emissions and mortality rates. Furthermore, lower energy consumption results in no fossil fuel mining and transportation demands, and more jobs will be created than will be lost during defossilization. All aforementioned global energy transition studies leading to very high shares of RE or 100% RE are based on energy system models. However, these are

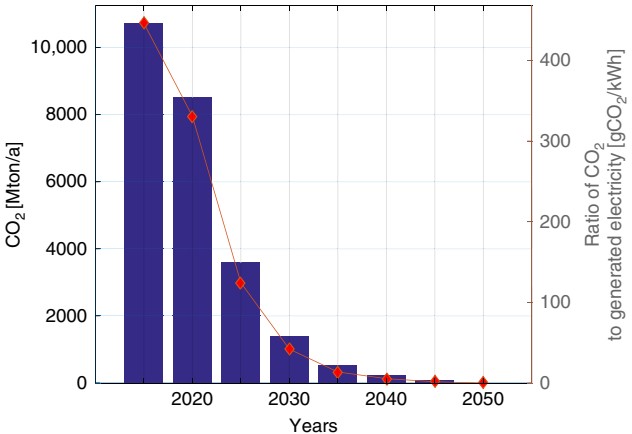

**Fig. 4** Global GHG emissions for the transition period 2015−2050. According to the existing trends in energy system development[50], the possible decrease of GHG emissions by 2020 will not be reached, global emissions may increase in comparison to the 2015 value. GHG greenhouse gas

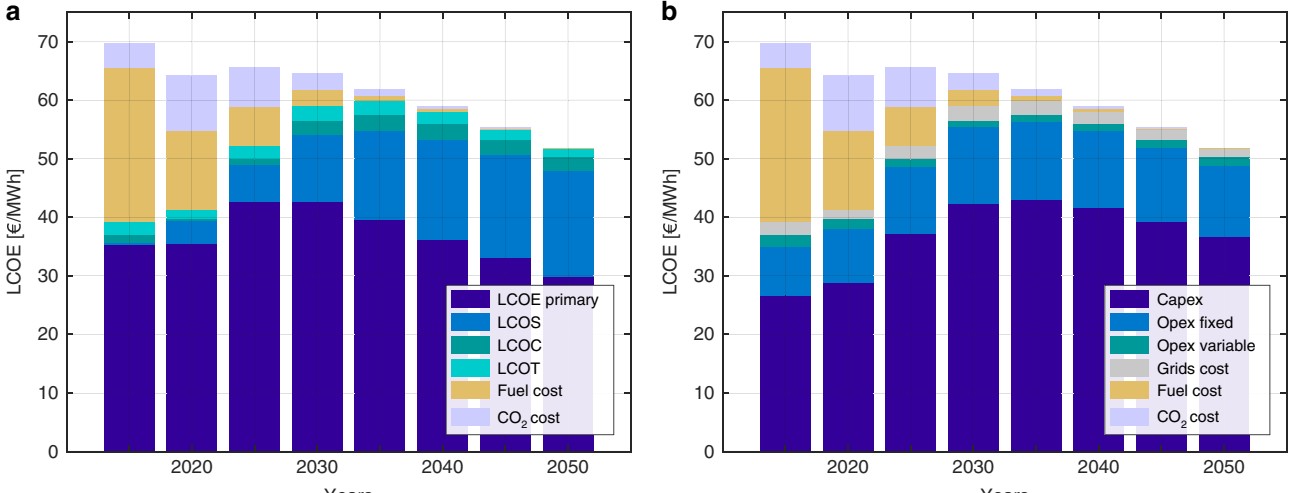

**Fig. 5** Globally averaged electricity system LCOE for the transition period from 2015 to 2050. LCOE primary levelized cost of electricity generation, LCOS levelized cost of storage, LCOC levelized cost of curtailment, LCOT levelized cost of transmission. **a** Breakdown by system components. **b** Breakdown by cost components. The energy transition leads to lower cost electricity supply. Applied financial and technical assumptions do not consider any breakthrough in efficiency or cost development, only evolutionary improvements and extending existing trends (see Supplementary Material for assumptions and results for all major regions)

**Table 1 Capital expenditures in the power sector for the transition from 2015 to 2050**

| Major region | Unit | Year | | | | | | |
|---|---|---|---|---|---|---|---|---|
| | | 2015–2020 | 2020–2025 | 2025–2030 | 2030–2035 | 2035–2040 | 2040–2045 | 2045–2050 |
| Europe | [b€] | 374–436 | 520–649 | 381–491 | 360–465 | 330–444 | 308–393 | 267–338 |
| Eurasia | [b€] | 89–95 | 266–302 | 76–92 | 65–78 | 48–62 | 48–62 | 75–96 |
| MENA | [b€] | 120–130 | 318–364 | 273–372 | 208–303 | 134–197 | 134–190 | 159–216 |
| SSA | [b€] | 50–55 | 133–156 | 125–175 | 119–168 | 119–169 | 161–232 | 205–287 |
| SAARC | [b€] | 150–163 | 342–415 | 545–771 | 362–514 | 341–493 | 404–581 | 438–623 |
| Northeast Asia | [b€] | 517–608 | 1223–1466 | 988–1340 | 693–988 | 619–885 | 693–1026 | 917–1278 |
| Southeast Asia | [b€] | 121–134 | 274–330 | 336–477 | 281–397 | 183–262 | 241–345 | 307–413 |
| North America | [b€] | 344–386 | 1126–1345 | 598–802 | 379–512 | 277–405 | 250–368 | 205–288 |
| South America | [b€] | 166–188 | 111–141 | 94–132 | 73–96 | 92–127 | 109–146 | 115–155 |
| Total | [b€] | 1920–2182 | 4307–5162 | 3415–4649 | 2538–3518 | 2140–3040 | 2347–3340 | 2679–3683 |

Including power generation, storage and interregional transmission. Capex numbers are given for 5-year periods including the uncertainty range, i.e. annual numbers would be roughly one fifth
*MENA* Middle East and North Africa, *SSA* sub-Saharan Africa, *SAARC* South Asian Association for Regional Cooperation

not yet in full hourly resolution and also limited in their spatial resolution. Integrated Assessment Models (IAMs) are very strong in linking the energy system to physical systems of Earth, and require more reduction in complexity in the energy system. IAM results in the IPCC AR5[3] showed low levels of RE, which has been criticized and at least partly traced back to very conservative cost assumptions, in particular for solar PV[20]. Recent results of IAMs have taken this criticism into account, in particular the too conservative solar PV cost assumptions. As a result, they now confirm very high shares of renewables[15]. However, there is not yet a 100% RE study carried out with IAMs. The IPCC SR1.5[10] provides an excellent overview of the latest results of ambitious pathway analyses towards the 1.5 °C target of the Paris Agreement and high RE shares are about 78% in the median, whereas the maximum share reaches 97%. Assumptions of pathways towards 1.5 °C for these IAMs do not vary widely compared to this research. However, IAMs still lack insights for storage needs, grid demand, demand response and VRE resource complementarity, since these models are typically operated using annual energy balancing, i.e. no temporal resolution. This research can answer some of these questions due to the full hourly resolution, in particular for storage and resource complementarity. Another criticism of IAMs and energy system models is that such models would be too normative[53] and not arbitrary in assumptions and results. A common weakness of techno-economic energy models is a lack of proper description of social dynamics and technology diffusion.

## Discussion

A global transition needs effort and investment, but each step can realistically lead to gradual, evolutionary change. A sustainable and carbon neutral electricity system based on 100% RE is technically feasible and economically viable globally by 2050 due to the reasonable total system LCOE (26–72 €/MWh) with a global average of 52 €/MWh (uncertainty range 45–58 €/MWh). Ongoing RE and storage cost decreases will position renewable electricity as the least cost source globally, and displace fossil fuel-based electricity, even with market mechanisms, unless the system is distorted by subsidies[54]. However, each regional energy transition will proceed rather uniquely. Each country will have a specific optimal electricity supply mix, but solar PV will become the dominating source of electricity globally. Beyond 2040, PV

will generate more than half of global electricity demand, and almost 70% in 2050. The 2020s will be most challenging due to the substitution of very high capacities of newly retired fossil fuel and nuclear capacities, and high capex. The transition will require a capex of around 22.5 trillion € (uncertainty range 19–25.5 trillion €), which is comparable to current power sector-related investments. Lifetime extensions of old fossil capacities and investments in new ones would result in additional challenges that complicate system development. For decades the RE share has grown slightly. However, despite discussions about defossilization and decarbonization of the energy system, GHG emissions keep on growing. In order to fulfill the Paris Agreement requirements as well as the United Nation's Sustainable Development Goals, a greatly accelerated transition should be started soon.

## Methods

**Modeling tool**. The transition modeling was performed with the LUT Energy System Transition model, which optimizes an energy system for given constraints. The simulation is applied for 5-year time periods for the years 2015−2050. For each period, the model defines a cost optimal energy system structure and operation mode for the given set of constraints: power demand, available generation and storage technologies, financial and technical assumptions, and limits on installed capacity for all applied technologies. The model is based on linear optimization and performed in an hourly resolution for an entire year (further details on the workings of the model along with the respective mathematical representation of the target functions can be found in Model section of Methods). The model ensures high precision computation and reliable results. The costs of the entire system are calculated as a sum of the annualized capital expenditures including the weighted average cost of capital (WACC), operational expenditures (including ramping costs), fuel costs and the cost of GHG emissions for all available technologies. The current model version is 2.0.

The LUT Energy System Transition modeling tool simulates and optimizes energy systems including the Power, Heat, and Transportation sectors, and additional Industry sectors, such as Industrial fuels production, Desalination and $CO_2$ removal. The simulation is performed in full hourly resolution for all hours of a year in single-year steps, where the starting conditions of the simulation depend on the time step assumptions and the previous time step results.

The purpose of the LUT Energy System Transition modeling tool is to assess different possible pathways of energy system development and assist global, national and regional energy strategy planning. Simulations allow investigation of the impact of different policies on the system structure, cost, emissions and the process of development. The model also tests the benefits of energy sectors integration (also called sector coupling), including the Power, Heat, Transportation and Industry sectors (for Industrial fuels production, Desalination and $CO_2$ removal), as well as evaluates the possibility of additional flexible demand option integration and its impact on the system. The model can be used for:

First, energy system development studies—simulation of the energy system transition from the current structure towards an optimized energy system: In such case, the simulation is performed for several time steps with specific financial and technical assumptions. The simulation starts from the existing energy system structure and the initial conditions of each time step are based on the system structure formed in previous steps. The results provide information on an optimized system structure and operation mode for each step, data on system cost, costs of all the products and elements, and GHG emissions of the system.

Second, feasibility studies—simulation of an optimized energy system structure and operation mode for the given technical and financial constraints: Instead of an energy transition, it is also possible to select an overnight approach, which can provide information on how a newly optimized energy system would look, built under given constraints.

Third, technical analysis—simulation of the system operation with given system structure, resource, technical and financial assumptions: Such simulations can be utilized for energy system robustness assessment to evaluate the range of conditions for which the system can satisfy the demand.

**Modeling procedure**. The first step of the energy system modeling is data preparation: defining the financial and technical assumptions. The structure of input data is described in the Input data section.

The second step is the scenario specification and simulation: available options are a transition scenario or overnight scenario. For each type of scenario, power, heat, transportation, and industry (industrial fuels production, desalination and $CO_2$ removal) sectors can be enabled. For the power sector, the simulation can be performed for a centralized system only or with the presence of power prosumers. For each type of simulation, three levels of regional integration can be applied: regional, country-wide, and area-wide. Regional: all regions (nodes) of the energy system are isolated. Country-wide: energy systems are integrated by transmission infrastructure, such as power grids, inside the same country. Area-wide: countries are integrated by transmission infrastructure for the selected area, typically a major region.

The third step is results preparation. After the end of the simulation, the tool collects the optimized results for all model elements in data files and summarizes the main data in a results Excel file. The description of the procedure and the structure of results file are given in the Results preparation section. The overall structure of the modeling procedure is given in Supplementary Fig. 52.

**Energy systems operation**. The model includes four energy sectors, each of which can also be simulated independently.

**Energy systems operation—Power sector**. The power sector is divided into a centralized energy system and a power prosumers subsegment. The share of electricity demand related to prosumers can be specified from 0 to 99% of total.

Centralized power system: In the centralized power system all consumption goes through the local AC grid to which the RE generation capacities (PV, wind, hydro, solar thermal electric, geothermal, biomass power plants), fossil and nuclear power plants, and fossil and biomass-based CHP plants are connected. At the same time, the local AC grid is connected to the storage capacities and interregional high voltage direct current (HVDC) and high voltage alternating current (HVAC) grids.

Power prosumers subsegment: PV prosumers represent three types: residential, commercial, and industrial. For each prosumer type, the share of total electricity demand (where the sum of residential, commercial and industrial is equal to the full power sector), grid electricity price, and financial assumptions for PV systems and batteries can be specified. Prosumers have the option to install their own PV generation capacities, Li-ion battery storage sell excess electricity to the centralized power system for a specified feed-in price or buy electricity from the centralized power system at a specified electricity cost. In the standard scenario the share of consumers willing to install their own PV generation capacities increases accordingly to a logistic function in steps of 3, 6, 9, 15, 18, and 20% of the respective segment electricity demand (if grid electricity is cheaper than that from PV generated, the share for the next step remains unchanged). If the power prosumer uses individual heating, generated power can also be used for electrical heating (heating rods and heat pumps). The simplified diagram of the power sector is presented in Supplementary Fig. 53.

**Energy systems operation—Heat sector**. The heat sector consists of six main segments: industrial high (>1150 °C), medium (100–1150 °C), and low (<100 °C) temperature heat demand, domestic water heating, space heating and cooking biomass demand. All heat shall be generated inside the region. The heat sector is also divided into centralized and individual heating systems.

All industrial heat must be covered by the centralized heat system, shares of centralized water and heating demand must be specified, and this must reflect the share of district heating specific for each region.

All biomass cooking, and the rest of water and heating demands are generated with individual heating systems.

The heat can be generated with CHP plants, solar thermal collectors, individual or centralized fuel-based boilers, electrical heaters, and heat pumps. Industrial high temperature heat demand can be satisfied only with fuel-based heat plants. Medium temperature heat can be also provided by electrical heating. Low temperature heat can also be satisfied by heat pumps, heating rods, solar thermal collectors and recovered heat loss from thermal power plants. Generated heat can be stored in medium or low temperature heat storage. The simplified diagram of the heat sector is presented in Supplementary Fig. 54.

**Energy systems operation—transportation sector**. The transportation sector is structured into the segments: road, rail, marine and aviation.

Within the road segment a separation is done for light duty vehicles, mainly cars; medium duty vehicles, such as delivery trucks; heavy duty vehicles; and buses. For the four road segments, the following powertrains are available: internal combustion engine, battery electric vehicle (BEV), hybrid plug-in vehicle (PHEV), and hydrogen-based fuel cell vehicles. The share of each type should be specified. BEVs and PHEVs are charged from the grid with "dump charge"—equally at every hour. Later model adjustments for "smart charge" and "vehicle-to-grid (V2G)" are planned.

Within the rail segment two fuel types are available: liquid hydrocarbon fuel (diesel), which can be fossil fuel, biofuel or renewable electricity-based Fischer-Tropsch (FT)-liquid fuel, and electricity. The shares of the fuels shall be selected according to respective projections.

Within the marine segment four fuel types are available: liquid hydrocarbon fuel (diesel), which can be fossil fuel, biofuel or renewable electricity-based FT-fuel; liquefied methane gas, which can be liquefied fossil natural gas, biomethane or renewable electricity-based methane (SNG); liquefied hydrogen (LH2), which is only foreseen as renewable electricity-based hydrogen, and electricity for shorter-distance domestic shipping.

Within the aviation segment three fuel types are available: liquid hydrocarbon fuel (kerosene), which can be fossil-based kerosene, biofuel or renewable electricity-based FT-kerosene; hydrogen, which is only foreseen as renewable electricity-based hydrogen; and electricity for shorter-distance flights.

The simplified diagram of the transportation sector is presented in Supplementary Fig. 55.

**Energy systems operation—industry sector**. The current model version includes the following industry sectors: industrial fuels production, desalination, and $CO_2$ removal. The inclusion of further industry sectors, such as cement, steel, chemical industry, metal refining and remaining industrial sectors, is planned for the future.

Industrial fuels production: The energy system can use fossil fuels, as long as it is allowed or affordable, convert biomass to biofuels, and produce renewable electricity-based synthetic fuels in the power, heat or transportation sectors. Currently hydrogen, methane and liquid hydrocarbons production units are integrated in the model.

Methane can be produced from biogas after its purification/upgrading. Then this biomethane can be used in the gas system. The share of biogas which can be upgraded is limited by the urbanization level of the region, but cannot exceed 70% even if the urbanization level is higher. A second option is synthetic natural gas (SNG)—methane produced with methanation reactors from hydrogen and carbon dioxide. The whole power-to-gas (PtG) system includes water electrolysis reactors (assumptions are based on alkaline technology) producing hydrogen from water, $CO_2$ direct air capturing (DAC) units collecting $CO_2$ and water from ambient air, and methanation units. Water electrolyzers and DAC units consume power from the system in order to produce $H_2$ and $CO_2$, and then methanation units convert them to synthetic $CH_4$.

Liquid hydrocarbons can be produced from biomass by biorefineries, or can be synthesized from $H_2$ and $CO_2$ using the FT process. PtG with gas storage and gas turbines can be part of storage for the power sector.

Fossil fuel refineries are not included in the model, and existing capacities of refineries are assumed sufficient to satisfy local consumption of fossil fuels.

The simplified diagram of the industrial fuels production sector is presented in Supplementary Fig. 56.

Desalination sector: Water demand in the region can be covered with Seawater Reverse Osmosis (SWRO) desalination, Multi-Stage Flash (MSF) and Multi-Effect Distillation (MED) technologies. The water is delivered to consumers by distributed piping systems with a respective energy demand, dependent on the distance and altitude from the coast. The water is stored at the production site, which may provide additional flexibility to the desalination system, and can optimize production in order to minimize total system cost. The simplified structure of the desalination sector is presented in Supplementary Fig. 57.

$CO_2$ removal sector: $CO_2$ removal demand can be specified for each region in tons of $CO_2$ per year. This amount of $CO_2$ will be captured from the atmosphere by DAC units in addition to $CO_2$ captured for synthetic fuels production. Heat and electricity needed for the DAC operation will be taken from the heat and power sectors, respectively. The simplified structure of the $CO_2$ removal sector is presented in Supplementary Fig. 58.

Integrated system: Every sector can be modeled individually or as several integrated sectors. Technologies such as PtG, electrical heating (heating rod, heat pumps), steam turbines, SWRO desalination, and FT-fuel production can operate as "bridging technologies" binding different sectors. Flexible power demand from the heat, transportation, industrial fuel production, desalination and

$CO_2$ removal sectors together with better energy management due to bridging technologies can lead to a significant increase in the integrated system efficiency and drop in the total system cost.

**Energy system elements**. All generation technologies are categorized into renewable-based, biomass-based, fossil-based power generation, renewable-based, biomass-based, fossil-based heat generation and fuel production technologies. Information on renewable-based power generation is summarized in Table 2. Information on biomass-based power generation is summarized in Table 3. Information on fossil-based power generation is summarized in Table 4. Information on renewable-based heat generation is summarized in Table 5. Information on biomass-based heat generation is summarized in Table 6. Information on fossil-based heat generation is summarized in Table 7. Information on fuel production technologies is summarized in Table 8.

All storage options can be divided into three main categories based on the typical energy-to-power ratio: diurnal (E/P ratio less than 24 h), mid-term storage (E/P ratio around 72 h), and long-term storage. Main information about storage technologies included in the model is summarized in Table 9.

Information on interregional power transmission technologies is summarized in Table 10. Information on water desalination and supply is summarized in Table 11.

**Model**. The energy system optimization model is based on a linear optimization of the system parameters under a set of applied constraints with the assumption of a perfect foresight of RE power generation and power demand. A multinode approach enables the description of any desired configuration of subregions and power transmission interconnections. The main constraints for the optimization are the matching of all types of generation and demand values for every hour of

---

**Table 2 Renewable-based power generation**

| Technology | Name | Abbr. | Inputs | Output | Additional |
|---|---|---|---|---|---|
| Solar PV | Utility-scale optimally tilted | RPVO | Min and max capacity limits | Optimal installed capacity | |
| | Utility-scale single-axis tracking (North-South) | RPVA | Capacity factors profile | Power generation profile | |
| | PV prosumers Residential | RPVR | | | |
| | PV prosumers Commercial | RPVC | | | |
| | PV prosumers Industrial | RPVI | | | |
| Wind turbines | Onshore Modern | RWIN | Min and max capacity limits | Optimal installed capacity | |
| | Onshore Old[a] | RWIO | Capacity factors profile | Power generation profile | |
| | Offshore Modern | ROWI | | | |
| Hydro | Run-of-river | RRRI | Min and max capacity limits | Optimal installed capacity | |
| | Reservoir (Dam) | HDAM | Capacity factors profile | Power generation profile | Average size of reservoir in days |
| Geothermal | Utility-scale power | TGEO | Min and max capacity limits | Optimal installed capacity | |
| | | | Hourly geothermal heat influx | Power generation profile | |
| Solar thermal | Utility-scale power | TSTU | Min and max capacity limits | Optimal installed capacity | |
| | | | Hourly CSP heat production | Power generation profile | |

[a]Modern onshore wind turbines have higher efficiency and respective higher capacity factors than old onshore wind turbines. All onshore wind turbines installed before 2015 are considered RWIO in order to avoid overestimation of existing turbine generation

---

**Table 3 Biomass-based power generation**

| Technology | Type | Abbr. | Fuel | Inputs | Output |
|---|---|---|---|---|---|
| Biomass | Power | TBPP | Biomass residues Biomass waste | Min and max capacity limits | Optimal installed capacity |
| | CHP | TCBP | Biomass residues Biomass waste | Energy conversion efficiency | Power and/or heat generation profile |
| Waste incinerator | CHP | TMSW | Municipal waste | Available amount of fuel | |
| Biogas CHP | CHP | TCHP | Biogas | | |

---

**Table 4 Fossil-fuel-based power generation**

| Technology | Type | Abbr. | Fuel | Inputs | Output |
|---|---|---|---|---|---|
| Gas | CCGT | TCCG | Natural Gas | Min and max capacity limits | Optimal installed capacity |
| | OCGT | TOCG | Biomethane | Energy conversion efficiency | Power and/or Heat generation profile |
| | CHP | TCNG | SNG | Available amount of fuel | |
| Coal | Power | THPP | Coal | | |
| | CHP | TCCO | | | |
| Liquid hydrocarbons | Power | TICG | Fossil liquids | | |
| | CHP | TCOI | biofuel | | |
| | | | FT-synfuel | | |
| Nuclear | Power | TNUC | Uranium | | |

**Table 5 Renewables/power-based heat generation**

| Technology | Type | Abbr. | Inputs | Output | Additional |
|---|---|---|---|---|---|
| Solar thermal | Utility-scale CSP | RCSP | Min and max capacity limits<br>DNI profile in [kWh/m$^2$] | Optimal installed capacity<br>Heat generation profile | |
| | Residential heat collector | RRSH | Min and max capacity limits<br>Capacity factors profile | | |
| Electrical heating | District heat | TDHR | Min and max capacity limits | Optimal installed capacity | |
| | Indiv. heat | THHR | Energy conversion efficiency | Heat generation profile | |
| Heat pump | District heat | TDHP | Min and max capacity limits | Optimal installed capacity | |
| | Indiv. heat | THHP | Energy conversion efficiency | Heat generation profile | |

**Table 6 Biomass-based heat generation**

| Technology | Type | Abbr. | Fuel | Inputs | Output |
|---|---|---|---|---|---|
| Biomass heat | District heat | TDBP | Biomass residues<br>Biomass waste | Min and max capacity limits<br>Energy conversion efficiency | Optimal installed capacity<br>Heat generation profile |
| Biomass heat | Indiv. heat | THBP | Biomass residues<br>Biomass waste | Available amount of fuel | |
| Biogas heat | Indiv. heat | THBG | Biogas | | |

**Table 7 Fossil-based heat generation**

| Technology | Type | Abbr. | Fuel | Inputs | Output |
|---|---|---|---|---|---|
| Gas | District heat | TDNG | Natural Gas | Min and max capacity limits | Optimal installed capacity |
| | Indiv. heat | THNG | Biomethane<br>SNG | Energy conversion efficiency<br>Available amount of fuel | Heat generation profile |
| Coal | District heat | TDCO | Coal | | |
| | Indiv. heat | THCO | | | |
| Liquid hydrocarbons | District heat | TDOI | Fossil liquids | | |
| | Indiv. heat | THOI | biofuel<br>FT-synfuel | | |

**Table 8 Fuel production**

| Name | Abbr. | Inflow | Outflow | Inputs | Output |
|---|---|---|---|---|---|
| Water electrolysis | TWEL | Power | Hydrogen | Min and max capacity limits<br>Energy conversion efficiency | Optimal installed capacity<br>Hydrogen generation profile |
| $CO_2$ DAC | TCOS | Power | Carbon dioxide | Min and max capacity limits<br>Energy demand for $CO_2$ production | Optimal installed capacity<br>$CO_2$ generation profile |
| Methanation reactor | TMET | Hydrogen<br>Carbon dioxide | Synthetic methane (SNG) | Min and max capacity limits<br>Feedstock demand for methane production | Optimal installed capacity<br>SNG generation profile |
| FT-reactor | TFTR | Hydrogen<br>Carbon dioxide | Liquid hydrocarbons<br>Naphtha | Min and max capacity limits<br>Feedstock demand for fuel production | Optimal installed capacity<br>FT fuel generation profile |
| Biorefinery | TBFR | Biomass<br>Power | Bio Liquid hydrocarbons | Min and max capacity limits<br>Feedstock demand for fuel production | Optimal installed capacity<br>Bio fuel generation profile |
| Biogas separator | TBGU | Biogas | Biomethane | Min and max capacity limits<br>Energy conversion efficiency | Optimal installed capacity<br>Biomethane inflow profile |
| Biogas digester | TBGD | biomass | Biogas | Min and max capacity limits | Optimal installed capacity<br>Biogas inflow profile |

the applied year, and the optimization criteria is the minimization of the total annual cost of the integrated system (or a sector if only a sector is optimized). The hourly resolution of the model significantly increases the required computation time; however, it guarantees that for every hour of the year the total supply within a subregion covers the local demand and enables a more precise system description including synergy effects of different system components or sectors (sector coupling).

The optimization is performed in a third-party solver. At the moment, the main option is MOSEK ver. 8, but other solvers (e.g. Gurobi, CPLEX, etc.) can also be used. The model is compiled in the Matlab environment in the LP file format, so that the model can be read by most of the available solvers. After the simulation results are parsed back to the Matlab data structure and can be postprocessed for analyses and diagram preparation.

**Model—target function**. The target of the system optimization is the minimization of the total annual cost of the integrated system (or a sector if only a sector is optimized), calculated as the sum of the annual costs of installed

**Table 9 Storage technologies**

| Name | Abbr. | Type | Inputs | Output |
|---|---|---|---|---|
| Utility-scale batteries | SBAT | Diurnal | Min and max capacity limits | Optimal installed capacity |
| Prosumer batteries (Residential) | SBAR | Diurnal | Charge and discharge Energy-to-power ratio | Charge and discharge profiles |
| Prosumer batteries (Commercial) | SBAC | Diurnal | Charge and discharge efficiency | |
| Prosumer batteries (Industrial) | SBAI | Diurnal | Self-discharge per hour | |
| Pumped hydro storage | SPHS | Diurnal | | |
| Hot heat storage | SHOT | Diurnal | | |
| Hydrogen storage | SHYD | Diurnal | | |
| District heat storage | SDHS | Mid-term | | |
| Biogas | SBGA | Mid-term | | |
| Adiabatic compressed air storage | SACA | Mid-term | | |
| Gas storage | SGAS | Seasonal | | |
| Liquid hydrocarbons | SLIQ | Seasonal | | |

**Table 10 Power transmission technologies**

| Name | Abbr. | Inputs | Output |
|---|---|---|---|
| HVAC Line | THAO[a] | Grid connections map | Optimal installed capacity of lines |
| HVDC Line | TRTL[a] | Min and max line capacity limits for connections | Energy flows profiles in both directions for lines |
| HVDC Converters station | TRCS | Efficiency | Power import exports profiles for regions |

*HVDC* high voltage direct current, *HVAC* high voltage alternating current
[a]HVAC and HVDC line financial assumptions should be calculated for the expected structure of the regional grid with average shares of underground cables and above-ground lines

capacities of the different technologies, costs of energy and product generation, and production ramping. This target function includes annual costs of the power, heat, transportation, and industrial (industrial fuels production, desalination and $CO_2$ removal) sectors. The target function of the applied energy model for minimizing annual costs is presented in Eq. (1) and comprises all hours of a year using the abbreviations: sub-regions ($r$, **reg**), generation, storage and transmission technologies ($t$, **tech**), capital expenditures for technology $t$ ($CAPEX_t$), capital recovery factor for technology $t$ ($crf_t$), fixed operational expenditures for technology $t$ ($OPEXfix_t$), variable operational expenditures technology $t$ ($OPEXvar_t$), installed capacity in the region $r$ of technology $t$ ($instCap_{t,r}$), annual generation by technology $t$ in region $r$ ($E_{gen\,t,r}$), cost of ramping of technology $t$ ($rampCost_t$) and sum of power ramping values during the year for the technology $t$ in the region $r$ ($totRamp_{t,r}$).

$$\min\left(\sum_{r=1}^{\mathbf{reg}}\sum_{t=1}^{\mathbf{tech}}(CAPEX_t \cdot crf_t + OPEXfix_t) \cdot instCap_{t,r} + OPEXvar_t \cdot E_{gen\,t,r}\right.$$
$$\left. + rampCost_t \cdot totRamp_{t,r}\right). \tag{1}$$

The power prosumers and individual heating users system are realized in an independent submodel with a slightly different target function. The prosumer system is optimized for each subregion independently, even if the subregion is connected to neighbors inside the area. The target function includes annual costs of the prosumer power generation and storage, heating equipment, the cost of electricity required from the distribution grid and the cost of fuels required for boilers. Income of electricity feed-in to the distribution grid is deducted from the total annual cost.

The target function of the applied energy model for minimizing annual costs is presented in Eq. (2) and comprises all hours of a year using the abbreviations: generation and storage technologies ($t$, **tech**), capital expenditures for technology $t$ ($CAPEX_t$), capital recovery factor for technology $t$ ($crf_t$), fixed operational expenditures for technology $t$ ($OPEXfix_t$), variable operational expenditures technology $t$ ($OPEXvar_t$), installed capacity of technology $t$ ($instCap_t$), annual generation by technology $t$ ($E_{gen\,t}$), retail price of electricity ($elCost$), feed-in price of electricity ($elFeedIn$), annual amount of electricity required from the grid ($E_{grid}$),

annual amount of electricity fed-in to the grid ($E_{curt}$).

$$\min\left(\sum_{t=1}^{\mathbf{tech}}(CAPEX_t \cdot crf_t + OPEXfix_t) \cdot instCap_t + OPEXvar_t \cdot E_{gen\,t}\right.$$
$$\left. + elCost \cdot E_{grid} + elFeedIn \cdot E_{curt}\right). \tag{2}$$

**Model—energy balance constraints.** The main constraint for the power sector optimization is the matching of the power generation and demand for every hour of the applied year as shown in Eq. (3). For every hour of the year the total generation within a subregion and electricity import cover the local electricity demand.

$$\forall h \in [1, 8760] \quad \sum_t^{\mathbf{tech}} E_{gen\,t} + \sum_r^{\mathbf{reg}} E_{imp\,r} + \sum_t^{\mathbf{stor}} E_{stordisch\,t}$$
$$= E_{demand} + \sum_r^{\mathbf{reg}} E_{exp\,r} + \sum_t^{\mathbf{stor}} E_{storch\,t} + E_{curt} + E_{other} \tag{3}$$

Eq. (3) describes constraints for the energy flows of a subregion. Abbreviations: hours ($h$), technology ($t$), all modeled power generation technologies (**tech**), subregion ($r$), all subregions (**reg**), electricity generation ($E_{gen}$), electricity import ($E_{imp}$), storage technologies (**stor**), electricity from discharging storage ($E_{stordisch}$), electricity demand ($E_{demand}$), electricity exported ($E_{exp}$), electricity for charging storage ($E_{storch}$), electricity consumed by other sectors (heat, transport, desalination, industrial fuels production, $CO_2$ removal) ($E_{other}$), curtailed excess energy ($E_{curt}$). The energy loss in the HVDC and HVAC transmission grids and energy storage technologies are considered in storage discharge and grid import value calculations.

The heat sector energy balance is defined by three equations: for industrial high temperature heat demand, for industrial high and medium temperature heat demand, and all centralized heat demand. High temperature heat can only be generated by fuel-based boilers (Eq. (4)). Medium temperature heat can also be generated by electrical heating and can be stored in high temperature heat storage and used to produce electricity with steam turbines (Eq. (5)). Low temperature heat can also be provided by heat pumps, electric heating rods and waste heat from other technologies (Eq. (6)).

$$\forall h \in [1, 8760] \quad \sum_t^{\mathbf{techHH}} E_{gen\,t} \geq E_{demandHH}, \tag{4}$$

$$\forall h \in [1, 8760] \quad \sum_t^{\mathbf{techHH}} E_{gen\,t} + \sum_t^{\mathbf{techMH}} E_{gen\,t} + E_{stordisch}$$
$$\geq E_{demandHH} + E_{demandMH} + E_{storch} + E_{other} \tag{5}$$

$$\forall h \in [1, 8760] \sum_t^{\mathbf{tech}} E_{gen\,t} + \sum_t^{\mathbf{stor}} E_{stordisch} = E_{demand} + \sum_t^{\mathbf{stor}} E_{storch} + E_{curt} + E_{other}. \tag{6}$$

Abbreviations: hours ($h$), technology ($t$), high temperature heat generation technologies (**techHH**), medium temperature heat generation technologies (**techMH**), all heat generation technologies (**tech**), industrial high temperature heat demand ($E_{demandHH}$), industrial medium temperature heat demand ($E_{demandMH}$), total centralized heat demand, including industrial, and space heating and water heating demand ($E_{demand}$).

Power and heat sector constraints for prosumers have some minor differences. Prosumers can buy electricity from electricity distribution companies

**Table 11 Water desalination and supply technologies**

| Name | Abbr. | Inflow | Outflow | Inputs | Output |
|------|-------|--------|---------|--------|--------|
| Reverse Osmosis Seawater Desalination | WROD | Power | Water | Min and max capacity limits<br>Desalination efficiency<br>Water demand profile | Optimal installed capacity<br>Water desalination profile |
| Multi-Stage Flash Stand alone | WMSS | Power<br>Heat | Water | Min and max capacity limits<br>Desalination efficiency<br>Water demand profile | Optimal installed capacity<br>Water desalination profile |
| Multi-Stage Flash Cogeneration | WMSC | Gas | Water<br>Power | Min and max capacity limits<br>Desalination efficiency<br>Water demand profile | Optimal installed capacity<br>Water desalination profile |
| Multi-Effect Distillation Stand alone | WMDS | Power<br>Heat | Water | Min and max capacity limits<br>Desalination efficiency<br>Water demand profile | Optimal installed capacity<br>Water desalination profile<br>Power |
| Multi-Effect Distillation Cogeneration | WMDC | Gas | Water<br>Power | Min and max capacity limits<br>Desalination efficiency<br>Water demand profile | Optimal installed capacity<br>Water desalination profile |
| Water storage | SWAT | Water | Water | Min and max capacity limits<br>Water demand profile | Optimal installed capacity<br>Charge and discharge profiles |
| Horizontal pumping | WHPU | Power | | Water demand profile | Optimal installed capacity<br>Pumping power demand profiles |
| Vertical pumping | WVPU | Power | | Water demand profile | Optimal installed capacity<br>Pumping power demand profiles |

(Eq. (7)). Heating of prosumers based on individual heaters includes fuel, RE and electricity-based heaters, but there is no individual heat storage option (Eq. (8)).

$$\forall h \in [1, 8760] \sum_{t}^{\text{tech}} E_{\text{gen } t} + \sum_{t}^{\text{stor}} E_{\text{stordisch}}$$
$$= E_{\text{demand}} - E_{\text{grid}} + \sum_{t}^{\text{stor}} E_{\text{storch}} + E_{\text{curt}} + E_{\text{other}}, \quad (7)$$

$$\forall h \in [1, 8760] \sum_{t}^{\text{tech}} E_{\text{gen } t} = E_{\text{demand}} + E_{\text{curt}}. \quad (8)$$

Abbreviations: hours ($h$), technology ($t$), all modeled power generation technologies (**tech**), energy generated ($E_{\text{gen}}$), storage technologies (**stor**), energy from discharging storage ($E_{\text{stordisch}}$), energy demand ($E_{\text{demand}}$), electricity energy for charging storage ($E_{\text{storch}}$), electricity consumed by heating ($E_{\text{other}}$), excess energy ($E_{\text{curt}}$).

**Model—power and heat generation**. The renewable-based power and heat generation is defined by historical capacity factors for this technology and the optimal installed capacity of this technology (Eq. (9)).

$$\forall h \in [1, 8760] \, E_{\text{genRE } h} = \text{CF}_{\text{genRE } h} \cdot \text{instCap}_{\text{genRE}}. \quad (9)$$

Abbreviations: hour ($h$), energy generated by renewable-based generation technology ($E_{\text{genRE}}$), capacity factor of the technology ($\text{CF}_{\text{genRE}}$), installed capacity in the region of the technology ($\text{instCap}_{\text{genRE}}$).

The fuel-based power and heat generation defined by the optimal installed capacity for this technology (Eq. (10)), availability factor for this technology (Eq. (11)), this technology used fuel available (Eq. (12)), and efficiency of the technology (Eq. (13)).

$$\forall \text{h} \in [1, 8760] E_{\text{genFU } h} \leq \text{instCap}_{\text{genFU}}, \quad (10)$$

$$\sum_{h}^{8760} E_{\text{genFU } h} \leq 8760 \cdot \text{AF}_{\text{genFU } h} \cdot \text{instCap}_{\text{genFU}}, \quad (11)$$

$$\sum_{h}^{8760} \text{FU}_{\text{genFU } h} \leq \text{totalFU}_{\text{genFU}}, \quad (12)$$

$$\forall h \in [1, 8760] E_{\text{genFU } h} = \text{FU}_{\text{genFU } h} \cdot \text{eff}_{\text{genFU}}. \quad (13)$$

Abbreviations: hour ($h$), energy generated by fuel-based generation technology ($E_{\text{genFU}}$), installed capacity in the region of the technology ($\text{instCap}_{\text{genFU}}$), availability factor of the technology ($\text{AF}_{\text{genFU}}$), fuel consumption for the hour $h$ ($\text{FU}_{\text{genFU } h}$), annual fuel consumption for the hour $h$ ($\text{totalFU}_{\text{genFU } h}$), energy conversion efficiency for technology ($\text{eff}_{\text{genFU}}$).

For all technologies, capacity is calculated in output units. For cogeneration the capacity is given in electrical units. For some types of fuel (municipal wastes, industrial biomass wastes, biogas) all available fuel must be consumed for sustainability reasons. Biogas inflow in the system is constant and biogas can be stored only for 48 h.

**Model—power and heat storage**. Storage technologies are described as energy storage capacity and storage interface capacity. Energy storage capacity limits the maximum state of charge (SoC) of the storage technology and the amount of energy stored (Eq. (14)), while the storage interface capacity limits the maximum power of charge and discharge (Eqs. (15) and (16)). The energy balance constraint for storage technologies is given in Eq. (17).

$$\forall h \in [1, 8760] \text{SoC}_{\text{stor } h} \leq \text{instCapEn}_{\text{stor}}, \quad (14)$$

$$\forall h \in [1, 8760] E_{\text{storch } h} \leq \text{instCapInt}_{\text{stor}}, \quad (15)$$

$$\forall h \in [1, 8760] E_{\text{stordisch } h} \leq \text{instCapInt}_{\text{stor}}, \quad (16)$$

$$\forall h \in [1, 8760] \text{SoC}_{\text{stor } h} = \text{SoC}_{\text{stor } h-1} \cdot \text{selfDisch}_{\text{stor}} + E_{\text{storch } h} \cdot \text{eff}_{\text{storch}} - E_{\text{stordisch } h} / \text{eff}_{\text{stordisch}}. \quad (17)$$

Abbreviations: hour ($h$), storage state of charge for an hour $h$ ($\text{SoC}_{\text{stor } h}$), installed energy capacity of the storage ($\text{instCapEn}_{\text{stor}}$), installed power capacity of the storage ($\text{instCapInt}_{\text{stor}}$), charging energy of the storage for an hour $h$ ($E_{\text{storch } h}$), discharging energy of the storage for an hour $h$ ($E_{\text{stordisch } h}$), hourly self discharge of the storage ($\text{selfDisch}_{\text{stor}}$), charge efficiency ($\text{eff}_{\text{storch}}$), discharge efficiency ($\text{eff}_{\text{stordisch}}$).

**Model—power transmission**. Power transmission is represented by HVDC and HVAC grids. Each line of the grid is bidirectional, but represented in the model as two unidirectional lines: import and export. Capacities of import and export lines are equal to the total power capacity of the interconnection, as shown in Eq. (18). Hourly export/import energy for a subregion is calculated as the sum of all import lines multiplied by this line transmission efficiency minus the sum of all export line energy flows, as shown in Eq. (19). The efficiency of energy transmission by HVDC lines depends on the distance and AC/DC converter pair efficiency, as shown in Eq. (20). The efficiency of energy transmission by HVAC line depends only on distance, as shown in Eq. (21). For both HVDC and HVAC the distance-related losses are calculated in a

simplified way.

$$\forall h \in [1, 8760] \text{line}_{\text{import}\,h} \leq \text{instCap}_{\text{line}}; \; \text{line}_{\text{export}\,h} \leq \text{instCap}_{\text{line}}, \quad (18)$$

$$\forall h \in [1, 8760] E_{\text{exp/imp}\,h} = \sum_{l}^{\text{lines}} \text{line}_{\text{import},l,h} \cdot \text{eff}_l - \sum_{l}^{\text{lines}} \text{line}_{\text{export},l,h}, \quad (19)$$

$$\text{eff}_l = \text{eff}_{\text{CS}} \cdot (1 - \text{distance} \cdot \text{EffLoss}), \quad (20)$$

$$\text{eff}_l = 1 - \text{distance} \cdot \text{EffLoss}. \quad (21)$$

Abbreviations: hour ($h$), line ($l$), energy flow through the power line (line), installed capacity of the power line (instCap$_{\text{line}}$), exported/imported energy for the region for an hour $h$ ($E_{\text{exp/imp},h}$), total energy import efficiency (eff$_l$), converter pair efficiency (eff$_{\text{CS}}$), charge length of the line (distance), energy loss in the line (EffLoss).

**Model—transportation.** Transportation demand is expressed in (metric) ton kilometers (t-km) and passenger kilometers (p-km). Power and fuel consumption for a given mix of transportation means is included in the power, heat and gas ($H_2$, $CH_4$) balance equations on the demand side.

**Model—industrial sector.** Fuel production: The energy system can produce GHG neutral methane for the needs of the power, heat, transportation and industry sectors. The first option is upgrading the available biogas to biomethane. The amount of upgraded biogas cannot be more than the urbanization level of the region, but not more than 70% of all biogas. Biomethane can be stored in the gas storage. The second option is power-to-gas. Hydrogen produced with water electrolysis and $CO_2$ from DAC units are used as raw materials for the methanation units. Produced SNG can be also stored in the gas storage.

Desalination: In case that desalinated water demand exists in the region, the system has to provide the demanded amount of water every hour. Water storage on the supply side provides flexibility to the system. Desalination units are located on the seashore and they can optimize work in order to decrease the total system cost. The water demand and water storage balance are described in Eqs. (22)–(23).

Water desalination units produce water and store it in water storage. Desalinated water production is limited by optimal capacities of enabled desalination plants and storage technologies (Eqs. (24)–(25)). Power, heat and gas consumption for desalination unit operation as shown in Eqs. (26)–(28) are included in the power, heat and gas balance equations on the demand side. The water pumping electricity demand according to Eq. (29) and cost is calculated based on the pumping capacity of the system, hourly water demand, weighted average length and head of the piping system.

$$\forall h \in [1, 8760] \sum_{t}^{\text{tech}} W_{\text{des}\,t,h} + W_{\text{stordisch}\,h} - W_{\text{storch}\,h} = W_{\text{demand}\,h}, \quad (22)$$

$$\forall h \in [1, 8760] \text{SoC}_{\text{stor}\,h} = \text{SoC}_{\text{stor}\,h-1} + W_{\text{storch}\,h} - W_{\text{stordisch}\,h}, \quad (23)$$

$$\forall h \in [1, 8760] W_{\text{des}\,t,h} \leq \text{instCapDes}_t, \quad (24)$$

$$\forall h \in [1, 8760] \text{SoC}_{\text{stor}\,h} \leq \text{instCapStor}, \quad (25)$$

$$\forall h \in [1, 8760] E_{\text{heat}\,h} = \sum_{t}^{\text{tech}} W_{\text{des}\,t,h} \cdot \text{heatCons}_t, \quad (26)$$

$$\forall h \in [1, 8760] E_{\text{el}\,h} = \sum_{t}^{\text{tech}} W_{\text{des}\,t,h} \cdot \text{elCons}_t - \sum_{t}^{\text{tech}} W_{\text{des}\,t,h} \cdot \text{elProd}_t, \quad (27)$$

$$\forall h \in [1, 8760] E_{\text{gas}\,h} = \sum_{t}^{\text{tech}} W_{\text{des}\,t,h} \cdot \text{gasCons}_t, \quad (28)$$

$$\forall h \in [1, 8760] E_{\text{elPump}\,h} = \sum_{t}^{\text{tech}} W_{\text{des}\,t,h} \times \left( \text{elCons}_{\text{VPump}} \cdot \text{alt} + \text{elCons}_{\text{HPump}} \cdot \text{dist} \right). \quad (29)$$

Abbreviations: hour ($h$), desalination technology ($t$), desalinated water ($W_{\text{des}}$), water storage discharge ($W_{\text{stordisch}}$), water storage charge ($W_{\text{storch}}$), water demand ($W_{\text{demand}}$), installed desalination technology capacity (instCapDes),

desalination heat demand ($E_{\text{heat}}$), desalination electricity demand ($E_{\text{el}}$), desalination gas demand ($E_{\text{gas}}$), desalination heat consumption (heatCons), desalination electricity consumption (elCons), desalination electricity production (elProd), desalination gas consumption (gasCons), water pumping electricity demand ($E_{\text{elPump}}$), horizontal water pumping electricity consumption (elCons$_{\text{HPump}}$), vertical water pumping electricity consumption (elCons$_{\text{VPump}}$), pumping distance (dist), pumping altitude difference (alt), water storage state of charge $h$ (SoC$_{\text{stor}}$), installed capacity of the water storage (instCapStor).

$CO_2$ removal: The energy system can capture additional amounts of $CO_2$ from the atmosphere for permanent storage. The $CO_2$ captured by DAC is stored in $CO_2$ buffer storage. The system will balance hourly DAC and $CO_2$ buffer operation in order to balance hourly $CO_2$ removal demand.

**Results preparation and cost calculations.** All optimization results are collected and converted from the solver output form to the Matlab structure. This structure contains all information about the system: installed capacities of all system elements, its operation modes, energy, fuel and other product flows.

Data on the structure and operation of the energy system in combination with financial and technical assumptions give the full description of the system. Based on these numbers, it is possible to calculate annual costs of each component and the whole system, allocate costs to specific sectors, calculate costs of products (electricity, heat, synthetic fuels, water) and different components of this costs (primary generation, storage, transmission, curtailment components of electricity prices etc.).

The total annualized cost of the system is calculated as the sum of all sectors costs (Eq. (30)), which includes annualized capital cost and operational costs of all system elements (Eq. (31)):

$$\text{totalCost}_{\text{sys}} = \text{elSysCost} + \text{elProsCost} + \text{heatSysCost} + \text{heatIndCost} \\ + \text{transpSysCost} + \text{industrSysCost}, \quad (30)$$

$$\text{totalCost}_{\text{sys}} = \sum_{t=1}^{\text{tech}} (\text{CAPEX}_t \cdot \text{crf}_t + \text{OPEXfix}_t) \cdot \text{Cap}_t + \text{OPEXvar}_t \cdot E_{\text{gen}\,t}, \quad (31)$$

$$\text{crf}_t = \frac{\text{WACC} \cdot (1 + \text{WACC})^{N_t}}{(1 + \text{WACC})^{N_t} - 1}. \quad (32)$$

Abbreviations: total annualized cost of the system (totalCost$_{\text{sys}}$), annualized cost of the centralized Power sector (elSysCost), annualized cost of the electricity prosumers sector (elProsCost), annualized cost of the centralized heat sector (heatSysCost), annualized cost of the individual heat sector (heatIndCost), annualized cost of the transportation sector (transpSysCost), annualized cost of the industrial sector (industrSysCost), all technologies (**tech**), technology ($t$), capital expenditures (CAPEX), capital recovery factor for technology $t$ (crf$_t$) Eq. (32), annual fixed operational expenditures (OPEXfix), variable operational expenditures (OPEXvar), installed capacity of the technology $t$ (Cap$_t$), annual output for the technology $t$ ($E_{\text{gen}\,t}$), weighted average cost of capital (WACC), lifetime for technology $t$ ($N_t$).

Total levelized cost of electricity in the system (LCOEtotal) is calculated as the electricity demand weighted average of the centralized power system LCOE (LCOEsys) and prosumers sector LCOE (LCOEpros); the formula is presented in Eq. (33). Centralized power system LCOE is comprised of levelized cost of consumed electricity (LCOEprim), levelized cost of storage (LCOS), levelized cost of curtailed electricity (LCOC), levelized cost of electricity transition (LCOT) and levelized cost of prosumer feed-in reimbursement (LCOFS), Eq. (34). For the prosumer sector, total LCOE is comprised of the levelized cost of consumed electricity (LCOEprim), levelized cost of storage (LCOS), and levelized cost of prosumer feed-in reimbursement (LCOFS), Eq. (35). Levelized cost of generated electricity is calculated as the total annualized cost of the electricity generation system divided by total annual generation (Eq. (36)). In these calculations, operational costs include costs of fuel and GHG emissions cost per unit of generated electricity. The electricity generation systems also include part of the fuel production facilities, which are used for fuel production for power system generators. Levelized cost of consumed electricity is calculated based on the cost of the generated electricity (LCOEgen), excluding electricity lost due to curtailment, storage and transmission system losses (Eq. (37)). Levelized cost of storage is calculated as the annualized cost of storage system equipment and annual cost of electricity losses divided by total electricity consumption (Eq. (38)). Storage systems also include part of the fuel production facilities, which are used for fuel production for the storage system generators (e.g. for power-to-gas−gas-to-power). Levelized cost of curtailment is calculated as the annual cost of curtailed electricity divided by total electricity consumption (Eq. (39)). Levelized cost of transmission is the calculated area total annualized cost of power grid equipment and annual cost of electricity losses divided by total electricity consumption, and multiplied by regional grid utilization weights (Eq. (40)), where regional grid utilization weights

are the average of regional shares of total export and import of energy (Eq. (41)).

$$LCOEtotal_r = (LCOEsys_r \cdot El_{consSys_r} + LCOEpros_r \cdot El_{consPros_r})/ \\ (El_{consSys_r} + El_{consPros_r}), \quad (33)$$

$$LCOEsys_r = LCOEprim_r + LCOS_r + LCOC_r + LCOT_r + LCOFS_r, \quad (34)$$

$$LCOEpros_r = LCOEprim_r + LCOS_r - LCOFS_r, \quad (35)$$

$$LCOEgen_r = \frac{\sum_{t=1}^{Gen}(CAPEX_t \cdot crf_t + OPEXfix_t) \cdot Cap_{t,r} + OPEXvar_t \cdot El_{gen,t,r}}{El_{gen,r}} \quad (36)$$

$$LCOEprim_r = \frac{LCOEgen_r \cdot (El_{gen,r} - El_{curt,r} - El_{storLoss,r} - El_{transLoss,r})}{El_{cons,r}}, \quad (37)$$

$$LCOS_r = \\ \left(\sum_{t=1}^{Stor}(CAPEX_t \cdot crf_t + OPEXfix_t) \cdot Cap_{t,r} + OPEXvar_t \cdot E_{out\,t,r} + LCOEgen_r \cdot El_{storLoss\,r}\right)/El_{cons\,r}, \quad (38)$$

$$LCOC_r = \frac{LCOEgen_r \cdot El_{curt\,r}}{El_{cons\,r}}, \quad (39)$$

$$LCOT_r = RegShare_r \cdot \left(\sum_{r}^{Reg}\sum_{t=1}^{trans}(CAPEX_t \cdot crf_t + OPEXfix_t) \cdot Cap_{t,r} + OPEXvar_t \right. \\ \left. \cdot El_{out\,t,r} + LCOEgen_r \cdot El_{transLoss\,r}\right)/El_{cons\,r}, \quad (40)$$

$$RegShare_r = 0.5 \cdot \frac{Import_r}{\sum_r Import_r} + 0.5 \cdot \frac{Export_r}{\sum_r Export_r}, \quad (41)$$

$$LCOFS_r = \frac{feedInTarif_r \cdot El_{prosTogrid\,r}}{El_{cons\,r}}. \quad (42)$$

Abbreviations: region (r), total levelized cost of electricity in the system (LCOEtotal), centralized system levelized cost of electricity (LCOEsys), prosumer sector levelized cost of electricity (LCOEpros), centralized system electricity consumption (El$_{consSys}$), prosumer sector electricity consumption (El$_{consPros}$), consumed electricity LCOE (LCOEprim), levelized cost of stored electricity (LCOS), levelized cost of curtailed electricity (LCOC), levelized cost of prosumer feed-in reimbursement (LCOFS), generated electricity LCOE (LCOEgen), power generation technologies (Gen), storage technologies (Stor), power transmission technologies (trans), technology (t), capital expenditures (CAPEX), capital recovery factor for technology t (crf$_t$), annual fixed operational expenditures (OPEXfix), variable operational expenditures (OPEXvar), installed capacity of the technology t (Cap$_t$), annual output for the technology t (El$_{gen\,t}$), annual electricity generation (El$_{gen}$), annual electricity curtailment (El$_{curt}$), annual storage loss (El$_{storLoss}$), annual grid loss (El$_{transLoss}$), annual electricity consumption (El$_{cons}$), annual output of storage t (E$_{out\,t}$), annual export of grid technology t (El$_{out\,t}$), electricity exported by region r (Export), electricity imported by region r (Import), feed-in reimbursement (feedInTarif), electricity sold by prosumers to the grid (El$_{prosTogrid}$).

The levelized cost of heat (LCOH) is calculated as the weighted average of the centralized and individual system LCOH (Eq. (43)). The centralized heat system LCOH (LCOHsys) and individual heat system LCOH (LCOHind) are calculated as the annualized cost of heat system equipment and annual cost of electricity consumption by heating equipment divided by total heat consumption (Eq. (44,45)). In both formulas, operational expenditures include the cost of fuel and GHG emissions per unit of generated heat. The heat systems also include part of the fuel production facilities, which are used for fuel production for heat generators. Cogeneration plants costs are only included in the power system.

Levelized cost of transportation (LCOM) is calculated as sum of the annualized cost of the entire transport fleet, cost of consumed fuel and electricity, GHG emission cost, divided by transportation demand (Eq. (46)).

Levelized cost of the industrial sector products (LCOP) are: levelized cost of gas (LCOG), liquid fuel (LCOF), water (LCOW), and CO$_2$ direct air capture (LCOD). These are calculated as the sum of annualized cost of the equipment and cost of annually consumed heat and electricity, divided by total annual consumption of the product (Eq. (47)).

$$LCOHtotal_r = (LCOHsys_r \cdot He_{consSys_r} + LCOHind_r \cdot He_{consInd_r})/ \\ (He_{consSys_r} + He_{consInd_r}), \quad (43)$$

$$LCOHsys_r = \left(\sum_{t=1}^{heat}(CAPEX_t \cdot crf_t + OPEXfix_t) \cdot Cap_{t,r} + OPEXvar_t \cdot He_{out\,t,r} \right. \\ \left. + LCOEsys_r \cdot El_{demSysHeat\,r}\right)/He_{consSys\,r}, \quad (44)$$

$$LCOHind_r = \left(\sum_{t=1}^{heat}(CAPEX_t \cdot crf_t + OPEXfix_t) \cdot Cap_{t,r} + OPEXvar_t \cdot He_{out\,t,r} \right. \\ \left. + ElPrice_r \cdot El_{demIndHeat\,r}\right)/He_{consInd\,r}, \quad (45)$$

$$LCOM_r = \frac{\sum_{t=1}^{Mob}(CAPEX_t \cdot crf_t + OPEXfix_t) \cdot Cap_{t,r} + FuPrice_{t,r} \cdot FuCons_{t,r}}{TR_{dem\,r}}, \quad (46)$$

$$LCOP_r = \left(\sum_{t=1}^{tech}(CAPEX_t \cdot crf_t + OPEXfix_t) \cdot Cap_{t,r} + OPEXvar_t \cdot Pr_{out\,t,r} \right. \\ \left. + LCOEsys_r \cdot El_{cons\,t,r} + LCOHsys_r \cdot He_{cons\,t,r}\right)/Pr_{cons\,r}. \quad (47)$$

Abbreviations: region (r), total levelized cost of heat in the system (LCOHtotal), centralized system levelized cost of heat (LCOEsys), individual heat sector levelized cost of heat (LCOHind), centralized system heat consumption (He$_{consSys}$), individual heat sector heat consumption (He$_{consPros}$), heat generation technologies (**heat**), transportation technologies (**Mob**), industrial sector production technologies (tech), technology (t), capital expenditures (CAPEX), capital recovery factor for technology t (crf$_t$), annual fixed operational expenditures (OPEXfix), variable operational expenditures (OPEXvar), installed capacity of the technology t (Cap$_t$), annual output for the technology t (He$_{out\,t}$), centralized system levelized cost of electricity (LCOEsys), retail price of electricity (ElPrice), electricity consumed by centralized heat system heaters (El$_{demSysHeat}$), electricity consumed by individual heat system heaters (El$_{demIndHeat}$), fuel price for transportation technology t (FuPrice$_t$), fuel consumption for transportation technology t (FuCons$_t$), transportation demand (TR$_{dem}$), annual product production (Pr$_{out}$), electricity consumption for the production (El$_{cons}$), annual heat consumption for the production (He$_{cons}$), annual product consumption (Pr$_{cons}$).

## Data availability

The data that support the findings of this study are available from the authors on reasonable request. The main model code is available from the authors on reasonable request.

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

## Acknowledgements

The authors gratefully acknowledge the public financing of Tekes for the "Neo-Carbon Energy" project under the number 40101/14 and support for Finnish Solar Revolution project under the number 880/31/2016 as well as support from the Energy Watch Group based on financing from Stiftung Mercator GmbH and Deutsche Bundesstiftung Umwelt. A.G. and A.S.O. would like to thank Fortum Foundation and LUT Foundation, for the valuable scholarship.

## Author contributions

D.B. was responsible for model and methodology development, simulation, results analysis, writing, and collected data for Eurasia and Northeast Asia major regions. J.F. collected the data on existing power capacities for all regions. K.S. collected data on transmission and distribution grid loss for all regions and estimated the values of future losses. A.A. collected data for MENA, North and South America major region, and was responsible for the preparation of diagrams, M.C. collected data for Europe major region, and gave support for paper writing, A.G. collected data for Southeast Asia and SAARC major regions, A.S.O. collected data for Sub-Saharan Africa major region, L.d.S.N.S.B. took part in South America data collection, C.B. analyzed the results, supervised, contributed to writing and coordinated the work.

## Additional information

**Competing interests:** The authors declare no competing interests.

