## [Peer Review File · Nature Communications]

Reviewers' comments:

Reviewer #1 (Remarks to the Author):

This paper is in the long tradition, starting with Amory Lovins in the mid-1970s, which sets out scenarios for transitions to low-carbon global energy systems. It is a serious energy systems analysis which appears to take a rather similar approach to many existing systems models (linear optimisation, including learning effects). The primary innovation appears to be the spatial disaggregation (145 regions), which allows national-level specification of low-C transitions. This is a useful improvement on previous work.

My main comments are:

1. Given the similarity to other energy systems and IAM assessments, it would be useful to have a fuller comparison of the assumptions and results of previous studies than the paper currently gives. There is a range of sources which have not been covered by the review presented here (the focus here is on Haegel and Creutzig).
2. The paper is a techno-economic analysis. One of the limitations of these studies is that they are frequently too optimistic about rates of adoption of new technology. At line 50, the authors states that 'all remaining challenges can be overcome'. Similarly at line 150, in discussing rates of diffusion of RETs, they state 'achieving this will be challenging but manageable'. Is it possible to elaborate on these conclusions? Other studies are not so sanguine.
3. The paper is primarily concerned with modelling supply of electricity. Can the authors say something about their modelling of demand, also in the context of changing structures of energy consumption?
4. One of the more surprising results in this scenario is the major role for gas storage (line 223 and Fig 3). Can the authors explain how they envisage the transition from fossil to biogas, including analysis of the major growth in biogas resources which they envisage?
5. More attention needs to be paid to some of the units in Figures, in particular the conflation of GHG and CO₂/CO₂-equivalent. Also, Figure 4 on global emissions probably needs to be updated to take account of published GHG emissions (see Global Carbon Project) - the trend shown for 2020 is now unlikely to be achieved.
6. The final sentence of the paper suggests that the lowC transition has still to start. I would argue that it has been underway for about 20 years!

Reviewer #2 (Remarks to the Author):

The article describes a pathway to 100 % renewable power production. The key results are the CAPEX and a global overview of the main sources of electricity.

The abstract and the article are well written. The authors reference previous literature appropriately. Analyzing all 145 regions was time-consuming.

However, I suggest rejecting the article for the following reasons:

- Results presented are not of immediate interest to many people because the described scenario is unlikely to happen
- I doubt the presented CAPEX. Besides the CAPEX, the results are missing novelty.

Further remarks:

- What about OPEX?
- Are there enough raw materials for so many PV cells?
- High costs for adapting the power grids are not considered.

I suggest submitting the article to another journal with a lower IF.

Reviewer #3 (Remarks to the Author):

The research paper is well focused and based on detailed techno-commercial assessment. It's interesting to see even imagination that entire world be sustaining on renewable energy.

But the key challenges on the potential of the resources, appropriate technologies, new technological interventions with respect to time, emerging clean energy technologies etc. are missing in the text. Further considering large hydro as fully renewable energy source is still a question of debate. There must be a section on energetics (energy embodied) and hence the claim on GHG emission reduction. Silicon based solar cells usually nit receives energy payback period in 20-15 years as their commercial project life. It would better if the resources (databases) could be mentioned in the paper for various renewable energy sources. Also, this need to specify that whether the energy generation numbers through various energy sources/ technologies are based on current parameters or any factor has been considered for enhancement of efficiencies in future. If yes than the rational need to be mention.

Beyond hydro most of the key renewables are highly intermittent in the nature (wind, solar – GHI, DNI); hence there will be huge impact on grid to accumulate and manage the variable energy generation. Will this be possible technically? Or there will be always buffer conventional power to manage the grid? How much commercial implications will be there to manage a grid with 100% renewable energy. Or will pumped hydro storage, conventional thermal power plants as buffer storage or batteries-based storage etc. will be the essential solutions? If batteries are being the anticipated solutions for grid integration and management issues than how this will suffice the GHG aspects. In longer run; is there any scope for Concentrating Solar Power technologies in longer run.

The estimated LCOE seems one representative data set; but this is effectively variable with respect to countries along with their inherent tax structures. How the generalized LCOE will be aligned with the bankability of such projects. Any recommendations on the project capacities in long run, bankability aspects of renewable energy projects could be useful for audience.

Reviewers' comments:

Reviewer #1 (Remarks to the Author):

This paper is in the long tradition, starting with Amory Lovins in the mid-1970s, which sets out scenarios for transitions to low-carbon global energy systems. It is a serious energy systems analysis which appears to take a rather similar approach to many existing systems models (linear optimisation, including learning effects). The primary innovation appears to be the spatial disaggregation (145 regions), which allows national-level specification of low-C transitions. This is a useful improvement on previous work. My main comments are:

1. Given the similarity to other energy systems and IAM assessments, it would be useful to have a fuller comparison of the assumptions and results of previous studies than the paper currently gives. There is a range of sources which have not been covered by the review presented here (the focus here is on Haegel and Creutzig).

Thanks for this very valuable point. We had such a section in the first manuscript, but had to remove it due to word limits for the submission. We are very grateful for your comment, so that we got the chance to include this most important section again in the manuscript. We have full coverage of all 100% RE studies on a global level included. In addition, some aspects of IAMs are mentioned, and linked to the relevant IPCC reports. The trend in IAM modelling is now also towards higher shares of renewables, but 100% RE studies are not yet published. The median 78% RE share in IAM results for the SR15 IPCC report is substantial progress compared to the previous AR5 IPCC report. This progress is acknowledged. General criticisms of IAMs, but also energy system models, are also incorporated in that part.

Additional text is added on Lines 377-412.

2. The paper is a techno-economic analysis. One of the limitations of these studies is that they are frequently too optimistic about rates of adoption of new technology. At line 50, the authors states that 'all remaining challenges can be overcome'. Similarly at line 150, in discussing rates of diffusion of RETs, they state 'achieving this will be challenging but manageable'. Is it possible to elaborate on these conclusions? Other studies are not so sanguine.

Most of the doubts about the feasibility of RE based energy systems are described in the Brown et al. paper [Brown]. Currently all necessary technologies are available on the market: RE generation technologies, short-term energy storage (pumped hydro and battery storage technologies) and even long-term storage. The costs of these technologies also show strongly decreasing trends with learning rates of about 15-20%. The remaining question is the stability of the grid, which tends to decrease with the growth of converter based demand and supply, and decrease of the rotary load [Inertia]. Denmark and Germany have shown that up to the current levels of variable renewables the grid is stable, in Germany even more stable than a decade earlier with less variable renewable. The inertia challenge can be solved with integration of synthetic inertia – basically improved algorithms of power converters. The case of South Australia is an excellent example that the newly added Tesla battery has substantially improved the grid stability and reduced the cost substantially, also due to the very fast reaction on a scale of 100 ms. That is why we expect that even these challenges can be overcome, and an RE-based system will not be less reliable in comparison to the current one. Fast growth of RE capacities is another

challenge on the way to an RE based system, but from the past, we see that PV module production can be scaled up very quickly. In recent years cumulative PV capacities in the US have been doubling every 2 years. Similar trends can be seen in China, India and other countries. Of course, much will depend on political will and social acceptance, but with the technical-economical approach we assume that political and social will follow the common good: cheap and sustainable energy supply. Additional text is added in Lines 53-72.

3. The paper is primarily concerned with modelling supply of electricity. Can the authors say something about their modelling of demand, also in the context of changing structures of energy consumption?

The power demand was modelled based on IEA demand growth projections (mainly from IEA World Energy Outlook 2015, which has not changed much in the meantime). These assumptions do not fully account for possible additional electricity demand in case of massive electrification of the heat and transport sectors. In this study we focus on development of the existing power sector and current structure of demand. Additional text has been added on Lines 154-161.

4. One of the more surprising results in this scenario is the major role for gas storage (line 223 and Fig 3). Can the authors explain how they envisage the transition from fossil to biogas, including analysis of the major growth in biogas resources which they envisage?

In this study we do not assume a major increase in biogas resources in the future. Our assumptions are based on data from European Commission (“Atlas of EU biomass potential”) and the German Biomass Research Center (“Regionale und globale räumliche Verteilung von Biomassepotenzialen”), and it is based on currently available sustainable resources for biogas production. We assume that utilisation of these valuable resources will grow and biogas production and utilisation will reach a maximum by 2030. In the first steps of the transition, biogas is mostly used locally in biogas CHP units, but with increasing RE share biogas becomes a valuable source of flexibility in the system. Biogas can be purified and stored in seasonal gas storage together with synthetic methane (SNG) produced using the PtG technology chain in order to compensate for the seasonal fluctuation of generation and demand. In total biogas resources can cover about one percent of electricity demand on a global scale. Thus, a 1% contribution to global electricity supply is noticeable, but still on a comparably low level.

Because of the seasonal storage application, gas storage capacity will be much bigger than the capacity of other, short to mid-term storage technologies, but from energy output perspective, its impact on the system is rather small, only 2% of electrical energy consumed will be stored in the gas storage, half contributed by biomethane and half by power-to-gas. As seasonal storage, the full charge cycles are only a fraction compared to diurnal storage, which explains large gas storage capacity but low storage output. In contrast, diurnal storage is characterised by very high storage output based on a comparably smaller energy storage capacity.

Additional text has been added on Lines 257-267.

5. More attention needs to be paid to some of the units in Figures, in particular the conflation of GHG and CO₂/CO₂-equivalent. Also, Figure 4 on global emissions probably needs to be updated to take account of published GHG emissions (see Global Carbon Project) - the trend shown for 2020 is now unlikely to be achieved.

Thank you for the comment. The units used in Figure 4 are changed to CO₂ equivalent. We account consistently CO₂ equivalent, which is however in the power sector dominated by CO₂. Yes, the pathway of transition discussed in the paper represents an optimal, technically possible decarbonisation process. Unfortunately, the energy system development shows inertia in decision-making and GHG emissions still grow in some parts of the world and are expected to grow in the near future.

6. The final sentence of the paper suggests that the lowC transition has still to start. I would argue that it has been underway for about 20 years!

You are right; RE has been an integral part of the energy system for decades and the share of RE is constantly growing, in particular since the 1990s and further accelerated in the 2000s by wind energy and in the 2010s by solar energy. However, fossil generation and GHG emissions are growing as well (as it is shown in the Global Carbon Project). We meant the real transformation of the system, which can break the trend of constantly growing GHG emissions, and decrease fossil fuel consumption. We have changed the sentence to emphasise this message.

Reviewer #2 (Remarks to the Author):

The article describes a pathway to 100 % renewable power production. The key results are the CAPEX and a global overview of the main sources of electricity. The abstract and the article are well written. The authors reference previous literature appropriately. Analyzing all 145 regions was time-consuming. However, I suggest rejecting the article for the following reasons:

- Results presented are not of immediate interest to many people because the described scenario is unlikely to happen

Currently more than 59 countries in the world have set a 100% RE target. These countries do not represent the major part of global economy; however, just recently California, the 5th largest economy accounted on its own, announced a 100% RE target. In recent weeks Spain and Portugal announced 100% RE targets for 2050. Uruguay demonstrated in recent years that the RE share can be increased from 60% to 97% while reducing cost. Countries, states and regions of such countries as Sweden, Germany, Spain, Italy, Denmark, Chile, and the United Kingdom have adopted their own 100% RE aims. So, for many countries, states and regions it is very important. Since the number of countries/ states making such decisions is growing, we support societal discourse on this important topic. IRENA showed in its recent 'Global Energy Transformation' report a 94% RE target for China and the EU and a 92% target for India by 2050, which represents the highest share of international institutions so far. We know that such international institutions typically follow recent scientific insights. Our manuscript also fulfils this demand. In addition, major global companies follow the dedicated target of 100% RE in all their activities globally to accelerate the process towards a fully sustainable energy supply, see <http://there100.org/companies>.

Since the 5th largest economy in the world, California, also home to many global leading technology innovations, also in the field of energy, is targeting 100% RE, we can assume that many people in the world are interested in how such a target can be fulfilled. This is also occurring in other parts of the world. The recent SR15 IPCC report further emphasised the urgent need for fast and massive decarbonisation.

- I doubt the presented CAPEX. Besides the CAPEX, the results are missing novelty.

The shown capital expenditures are truly one of the most important results of the simulation, showing that the transition towards 100% renewables is possible and affordable: the capital investments in RE generation, storage and additional power transmission capacities will be at a level comparable to the conventional power system. To avoid any possible doubts about these numbers, we have shown capex assumptions for all assessed technologies (Table S5) and the projected installed capacity numbers for all these technologies and major regions (Tables S9, S11, S13, S15, S17, S19, S21, S23, S25, S27). With these data it is possible to reproduce the shown capital expenditures with high precision (minor deviations are possible because pre-2015 values are not given). Data for global and regional power generation (Tables S10, S12, S14, S16, S18, S20, S22, S24, S26, S28) shows that such an energy system will be able to satisfy growing electricity demand (Table S4). We spent much effort to be as transparent as possible with our assumptions and results for the most important matter of capex.

The capex numbers for all technologies (Table S5) are based on references, which are also shown in the supplementary materials. The most important ones related to solar PV, which are based on the consensus view of leading European solar PV experts working in science, administration and industry.

Current PV capex values are lower in leading countries in the world, as we have assumed, and as published recently by leading global PV players in their recent ITRPV report (<http://itrpv.net/Reports/Downloads/>). Battery capex values are based on the latest industry data and learning rates, published in Nature Energy (see our references Schmidt et al. and Kittner et al.) – but we are even more conservative, since by full application of the learning rates the battery capex would be even lower.

PV and battery represent the highest fraction in total CAPEX, which is the sum of all single positions.

Results represent novelty, since there is not a single article for the energy transition towards full sustainability in the power sector in such high temporal and spatial resolution. In addition, we report in detail on the respective economics, which is also not published in a single article in that global-local resolution. We cooperate closely with all leading research groups for fully sustainable energy systems in the world, such as Henrik Lund from Aalborg University (also Editor-in-Chief for 'Energy') and co-author of the Brown et al. article in Renewable and Sustainable Energy Reviews, and we have a common database of all the nearly 200 journal articles on fully sustainable energy systems. In all these publications, however, such an article as ours is not recorded.

Further remarks:

- What about OPEX?

Opex variable and Opex fixed assumptions for all technologies during the transition period 2015-2050 are given in the the Supplementary Materials (Table S5). Annual operational costs of the system are not shown directly, but it is one of the integral elements of LCOE (levelized cost of electricity) calculations and its impact on LCOE for different regions during the transition period is shown in Figures S29-S37.

The impact of the transition on operational expenditures is briefly discussed after Figure 5.

- Are there enough raw materials for so many PV cells?

That depends on the technology. For CdTe thin-film PV modules, the resource base is very limited, but this technology contributes less than 5% of the total annually installed capacity, and in principle it would not be required at all. For silicon-based PV, representing more than 95% of the annually added solar PV capacity, main raw materials from a mass content point of view are silicon (for glass and semiconductor material) and aluminium, two of most abundant elements in the Earth's crust. Mass content of doping materials is negligible. Silicon solar cells often use silver, but this is not mandatory, as documented by the high-efficiency PV cells of SunPower. The specific silver demand per Wp of PV capacity has been shrinking for years, due to the continued improvement in the technology. Silver, used as metal contact to extract charge carriers from the solar cell, can be also substituted by copper, which is substantially less limited than silver and can be itself substituted by aluminium for major copper demand, e.g. for power lines. Aluminium is not practically limited. Some foils are needed in addition for solar PV modules, but their chemistry is based on hydrocarbons, for which we are also not limited, and even fossil-free hydrocarbon production routes could be used. In total, there is no material limitation known to produce

these capacities of PV, but precise calculations of that are a study on its own. Thanks for your implicit recommendation for preparing such a study in future.

- High costs for adapting the power grids are not considered.

Costs of adapting and reinforcement of transmission and distribution grid for the high share of RE systems was discussed in detail in the earlier mentioned Brown et al. article. Even in a highly centralised power system, the cost of distribution grid reinforcements will add 10-15% to the final cost of electricity. In highly decentralised systems, this value will be even lower, because in a prosumer based system, the role of distribution grid will decrease. The transmission and distribution grid losses also decrease over time, as recently published in an article of Sadovskaia et al. (<https://doi.org/10.1016/j.ijepes.2018.11.012>).

The cost of distribution grid reinforcements is discussed in the text after Table 1.

I suggest submitting the article to another journal with a lower IF.

We find it difficult to respond to this comment, as we have taken the decision to publish in this journal very seriously. We see the reviewer acknowledge that the manuscript is worthy of publication, but just not here. With all due respect to the reviewer's personal opinion, we disagree, and are firmly committed to making this manuscript appropriate for this journal.

Reviewer #3 (Remarks to the Author):

The research paper is well focused and based on detailed techno-commercial assessment. It's interesting to see even imagination that entire world be sustaining on renewable energy.

But the key challenges on the potential of the resources, appropriate technologies, new technological interventions with respect to time, emerging clean energy technologies etc. are missing in the text.

The power system modelling results presented in this paper show that the available RE resources are adequate to supply the growing power demand of the future. The results show an optimal mix of technologies necessary to balance energy systems for various regional conditions. In this study we only use existing technologies and assume their evolutionary development, without any significant breakthrough.

Our main aim is to show the possibility of a transition towards 100% RE supply with only existing technologies, assuming its evolutionary development without any significant breakthrough. That is the reason why we avoid discussions of possible new inventions and their impacts on the possible trajectory of development. In fact, all additional breakthroughs and further inventions could further improve the available technology base and thus accelerate the transition towards 100% RE.

Raw material demand for the RE capacity manufacturing is research on its own. Preliminary results of our colleagues from DLR show that Lithium and Dysprosium can be limiting factors for our scenario¹⁾, if only Li-ion batteries and permanent magnet generators of wind turbines are considered. Fortunately, the current range of competing manufacturing technologies of batteries (which do not demand Lithium) and wind turbines generators (without permanent magnets and thus without dysprosium and neodymium) exist on market.

1) [https://elib.dlr.de/120431/1/Junne%20et%20al.. Material%20Demand%20in%20Global%20Energy%20Scenarios ESCC%20Conference FIN.pdf](https://elib.dlr.de/120431/1/Junne%20et%20al..%20Material%20Demand%20in%20Global%20Energy%20Scenarios%20ESCC%20Conference%20FIN.pdf)

Discussion on raw materials scarcity has been added on Lines 62-72.

Further considering large hydro as fully renewable energy source is still a question of debate. There must be a section on energetics (energy embodied) and hence the claim on GHG emission reduction.

In our scenario, we do not assume massive commissioning of new hydropower plants, only up to an additional 50% of capacity can be added to the system, which reflects commissioning of currently under-construction plants and modernisation of existing hydro capacities. In 2015, about 1028.5 GW of hydropower capacities were in operation and about 185 GW were under construction globally (accordingly to the GlobalData database). In our scenario, by 2050 cumulative hydropower capacities have increased only by 253 GW. The main reason to have these strict constraints is the negative impact on river ecosystems, and substantial CO₂ emissions during cement and steel production for hydro dams also have substantial negative impacts. We mainly accept the reality of existing plants and plants under construction. In other research we even show that massive new hydropower capacities are not needed, due to sustainability and cost reasons (doi:10.3390/en11040972). Since massive commissioning of hydropower plants is not part of the scenario, and mostly built or under construction hydropower plants are considered, we do not see the reason to add the section about hydro power plant energetics in the paper.

Silicon based solar cells usually nit receives energy payback period in 20-15 years as their commercial project life. It would better if the resources (databases) could be mentioned in the paper for various renewable energy sources. Also, this need to specify that whether the energy generation numbers through various energy sources/ technologies are based on current parameters or any factor has been considered for enhancement of efficiencies in future. If yes than the rational need to be mention.

The Energy Payback Time for silicon-based solar cells is considerably less than 2 years, as documented in numerous research papers in recent years. The technical lifetime is 25-30 years right now, which documents the high and sustainable value of this technology. We have added two references for this, one in the leading solar PV journal providing an outlook for 2020 and authored by major researchers in this specific field (they report an energy payback of around 1 year), and another referring to more than 100 papers on the topic and presenting the first ever reported energetic learning curve for solar PV, which describes well the continuously decreasing Energy Payback Time.

We are much more worried about the Energy Payback Time of energy crops, since there the payback period is in the order of the technical lifetime, so that we do not assume energy crops in the simulation, but bioenergy based on by-products and residues.

Data sources for RE resource availability and the actual resource availability assumptions are presented in Supplementary Materials (Table S2). Also, we show our assumptions on maximum installed capacity limits for all regions (Table S3). Capacity factors and area requirements for PV and wind turbines are calculated for the currently widely available technology: PV systems with 15% efficiency, which is conservative given the fact that current PV modules typically have around 18% efficiency, and 3 MW Enercon E-101 turbines. Biomass available resources are based on current assumptions by DBFZ and the European Commission. However, we assume progress in energy conversion technologies such as combined cycle gas turbines, biomass power plants, and waste incinerators. Efficiency assumptions for all conversion technologies are presented in the Supplementary Materials (Table S5). For most conventional technologies, we refer to assumptions of the European Commission.

Beyond hydro most of the key renewables are highly intermittent in the nature (wind, solar – GHI, DNI); hence there will be huge impact on grid to accumulate and manage the variable energy generation. Will this be possible technically?

The system is modelled in full hourly resolution, and the main constraint of the model is the balance of power supply and demand for every hour of the year. To avoiding huge copper plate effects, we also model the world in 145 regions, which is far beyond the state of the art of around 20 regions for global models. Model structure and constraints are better described in the Supplementary Materials. As is discussed in the ‘Radical transition in evolutionary steps’ section, growth of RE generation share will result in an increase in storage capacities. The storage technologies will play a most important role in the balancing of supply and demand, except in the countries with high availability of hydro resources and high shares of hydro dams, like the Pamir region. There hydro can operate as virtual storage and balance high shares of intermittent renewables. This has been published recently in another article (<https://www.sciencedirect.com/science/article/pii/S1876610218309858>). Grids always have a balancing function, and they are more important in big countries with integrated power systems and

regions with excellent wind resources, but even there spatial coupling of storage and generation capacities will decrease needed grid interconnection capacity.

Please do not forget the very important balancing function of bioenergy. Hydropower and bioenergy are both dispatchable renewables and they will be used more with this feature in the future. This is because solar and wind both emerge as the least cost sources of bulk electricity, and dispatchable renewables are lower in cost than most storage options, in particular seasonal storage.

The article of Brown et al. shows that in the most affected grids in the world right now, the management of variable renewable energy can be well organised, such as in Denmark, Germany or California.

Or there will be always buffer conventional power to manage the grid? How much commercial implications will be there to manage a grid with 100% renewable energy. Or will pumped hydro storage, conventional thermal power plants as buffer storage or batteries-based storage etc. will be the essential solutions?

As is described in 'Radical transition in evolutionary steps' section, the system can operate with 100% RE generation (VRE and dispatchable/ flexible hydro dam and biomass power plants) with support of storage technologies, without the need for conventional power plants as a buffer. Sufficient storage capacity will guarantee the system stability, since the current 'rotating mass reserve' can be substituted by synthetic inertia appliances. There will be still conventional gas turbines, as described in the manuscript, but the fossil fuel is substituted by chemically identical biomethane and power-to-gas. The existing pumped hydro energy storage capacities are used as well as much more highly flexible battery systems, which can provide various grid services, as currently demonstrated in various places around the world, most prominently in South Australia, where the Tesla battery reduces the cost and increases grid stability. As described in the manuscript, batteries will play a most important role in storing electricity and providing grid services.

If batteries are being the anticipated solutions for grid integration and management issues than how this will suffice the GHG aspects. In longer run; is there any scope for Concentrating Solar Power technologies in longer run.

Battery storage does not produce any GHG emissions in operation. Currently, batteries have a carbon footprint related to the energy needed to manufacture them, but with an increase of the RE share in generation, this will shrink to zero, since the used energy will be also carbon free. In addition, recycled batteries show a better carbon footprint than initially produced ones. We clearly show how the power sector can be brought to zero GHG emissions. At that time, electricity used for battery manufacturing is also based on this GHG emission level.

The cost benchmark of PV-battery is very competitive for concentrating solar thermal power. To our best knowledge, based on the applied cost assumptions, the outcome shows no bright future for CSP. This is also confirmed by current market trends, since almost no new CSP capacity have been added in the last 3 years, whereas in solar PV the current global installations in 3 weeks exceed the historic cumulative installed CSP capacity. If there would be technological/ economic breakthroughs for CSP which we cannot anticipate, then the situation may change. So, it is possible. However, in the model CSP was one of available technologies for system, but with current and projected technical and financial assumptions, based on the latest insights of the academic leader in CSP research, the German Aerospace

Center (DLR), it was not competitive in comparison to others. CSP may have a chance as a high temperature heat option; however, this was not part of this manuscript, since we discuss the transition in the power sector.

Concentrating solar PV power had been squeezed out of the market in recent years. Whether there will be new breakthroughs that will cause CPV to come back, we cannot anticipate, but it is possible. The former CPV world market leader, Soitec/ Concentrix, had the largest installed capacity base, best academic backing, highest efficiency world records, and an efficient management team. However, they could not withstand the very fast cost decline of conventional c-Si PV technology. Even worse, no single investor could be found to re-invest in the bankrupt company, so that all assets were liquidated. No new players of relevance have re-started CPV in recent years.

The estimated LCOE seems one representative data set; but this is effectively variable with respect to countries along with their inherent tax structures. How the generalized LCOE will be aligned with the bankability of such projects. Any recommendations on the project capacities in long run, bankability aspects of renewable energy projects could be useful for audience.

Our intention had been to show the cost without subsidies and without taxes. How countries push the transition with taxes/subsidies is out of the scope of this research, but in general higher taxes for GHG emissions would accelerate the transition and higher support for RE would also push it. In contrast, subsidies for fossil fuels (as practiced today) would slow down the transition and no GHG emission price would also slow down the transition. However, we already know that some countries still would have a very strong rationale to go for very high RE shares (95% or more), as we have already shown for the case of Sun Belt countries (e.g. Israel (<https://doi.org/10.1016/j.energy.2018.05.014>), or Nigeria (<https://doi.org/10.1016/j.enconman.2018.10.036>)).

Reviewers' comments:

Reviewer #1 (Remarks to the Author):

I am satisfied that the authors have responded adequately to the comments of the 3 reviewers. They have been able to provide additional technical detail on a wide variety of issues raised by the reviewers, both in the text and in supplementary materials.

The paper contains some useful new insights about how a zero-C global emissions scenario can be achieved. As they point out, there have been relatively few 100% RE scenarios published and the LUT model provides an additional piece of evidence for policymakers and analysts who may be interested in pursuing this goal. The paper draws on a number of interesting methodological innovations.

I would recommend publication.

Reviewer #2 (Remarks to the Author):

Thank you for the detailed response to my review.

First of all, I do acknowledge the manuscript is worthy of publication. In particular, the appendix contains many interesting information (e.g. detailed comparison between the regions, when does PtG becomes relevant, what energy mix is relevant for the different regions and transition steps ...). Unfortunately, the main text is focused on the calculated costs. I still doubt the presented LCOE for the following reasons:

- Such a complex case is practically impossible to calculate. Although exact numbers are presented. A price range would be preferable.
- In engineering, a budget for unforeseeable costs is assumed. This is missing.
- Many assumptions are required for the calculation. A sensitivity analysis is helpful to identify the more important assumptions.

Two examples for the assumptions:

The reference for the costs of "Wind onshore" (appendix, ref 15) is more than 10 years old. The CAPEX is relatively high (< 1000 €/kW is more realistic for 2020).

The assumptions for “water electrolysis” are very optimistic (e.g. efficiency of 85 % for 2015). For 2020, a CAPEX of 685 €/kW is assumed. In contrast, a study of E&E Consultant assumes 1250 €/kW in 2020.

In case of an acceptance:

I recommend a sentence about the uncertainties of the cost estimation before publishing.

Reviewer #3 (Remarks to the Author):

Recommended for publication.

Reviewers' comments:

Reviewer #1 (Remarks to the Author):

I am satisfied that the authors have responded adequately to the comments of the 3 reviewers. They have been able to provide additional technical detail on a wide variety of issues raised by the reviewers, both in the text and in supplementary materials.

The paper contains some useful new insights about how a zero-C global emissions scenario can be achieved. As they point out, there have been relatively few 100% RE scenarios published and the LUT model provides an additional piece of evidence for policymakers and analysts who may be interested in pursuing this goal. The paper draws on a number of interesting methodological innovations.

I would recommend publication.

Thank you for your very positive and encouraging feedback.

Reviewer #2 (Remarks to the Author):

Thank you for the detailed response to my review.

First of all, I do acknowledge the manuscript is worthy of publication. In particular, the appendix contains many interesting information (e.g. detailed comparison between the regions, when does PtG becomes relevant, what energy mix is relevant for the different regions and transition steps ...). Unfortunately, the main text is focused on the calculated costs. I still doubt the presented LCOE for the following reasons:

- Such a complex case is practically impossible to calculate. Although exact numbers are presented. A price range would be preferable.

Thank you for your comment. The numbers were calculated for the presented system structure and given technical and financial assumptions. Technically, with these data it is possible to calculate the cost of such a system. However, uncertainty in the given cost and technical assumptions obviously exists. We have added discussion on the cost uncertainty and its impact on the final cost of the system, as well as a range of values for LCOE and capital expenditures estimations.

- In engineering, a budget for unforeseeable costs is assumed. This is missing.

The statement about unforeseeable cost is added to the discussion on cost uncertainty.

Please have in mind:

Current PV capex values are lower in leading countries in the world than we have assumed, and as published recently by leading global PV players in their recent ITRPV report (<http://itrpv.net/Reports/Downloads/>). The PV fixed-titled system capex for 2020 are assumed to be below 485 €/kW (at 1.3 USD/€) by ITRPV, in contrast to that our cost assumptions for 2020 are 580 €/kW. This is a considerably more conservative assumption than the current state of the industry. Battery capex values are based on the latest industry data and learning rates, published in Nature Energy (see our references Schmidt et al. and Kittner et al.) – but we are even more conservative, since by full application of the learning rates the battery capex would be even lower (about 30 €/kWh, cap of battery capacity, but we assume 75 €/kWh, cap).

It may be justified to turn the question upside down, what may be an additional benefit from faster than expected cost reductions.

In addition we would like to point out that such a budget for unforeseeable costs is already largely included in an additional position for many regions in the world, since the chosen WACC of 7% is much higher than currently required in main markets. A good example is the case of Germany, where a recent Nature Energy publication (<https://doi.org/10.1038/s41560-018-0277-y>) concluded WACC of 2.5% for solar PV and 2.75% for wind (70% debt, 30% equity) for recent projects. This is well below the chosen 7%, which is a kind of standard in energy system modelling and represents an additional capex budget of about 50-65% (depending on the lifetime assumptions for the PV systems).

- Many assumptions are required for the calculation. A sensitivity analysis is helpful to identify the more important assumptions.

Two examples for the assumptions:

The reference for the costs of “Wind onshore” (appendix, ref 15) is more than 10 years old. The CAPEX is relatively high (< 1000 €/kW is more realistic for 2020).

The assumptions for “water electrolysis” are very optimistic (e.g. efficiency of 85 % for 2015). For 2020, a CAPEX of 685 €/kW is assumed. In contrast, a study of E&E Consultant assumes 1250 €/kW in 2020.

Thank you for these comments. A sensitivity analysis was performed in order to define the actual impact of core system technologies (PV, wind power plants, Batteries, Power-to-Gas) on the cost of the system. Text is added to the manuscript.

You are right, water electrolyzers efficiency for the first steps of transition is optimistic, however it does not have significant impact on the system cost until late 2030s, when Power-to-Gas emerges. By that time efficiency of electrolyzers is expected to reach this level. Capex of large-scale electrolyzers can be even lower than mentioned 685 €/kW_{el} in 2020, according to IEA it can be on the level of 350 €/kW_{el} (450 USD/kW_{el}, exchange rate 1.3 USD/€) for a unit with 70% efficiency) [https://www.iea.org/publications/insights/insightpublications/Renewable_Energy_for_Industry.pdf].

The wind capex seems to be well balanced, since the largest cost reduction was from better wind yield due to better adapted wind turbines. This improvement is factored in by choosing a respective wind turbine type for the yield modelling (E-101 from Enercon) and allowing modern wind turbine hub heights. However, the wind power capex has not much improved compared to our assumed values. In case you know better scientific references for wind power plant capex (not only turbines, the full system including grid connection), showing that our assumptions are too conservative, please let us know. We would be happy work that in. We do regular literature reviews, but have not found relevant literature which would justify lower wind power plant capex values. The positive learning curve for wind LCOE is mainly due to the improvements in increasing the yield.

In case of an acceptance:

I recommend a sentence about the uncertainties of the cost estimation before publishing.

Reviewer #3 (Remarks to the Author):

Recommended for publication.

Thanks for the very positive feedback.